# Parameterizing microphysical effects on variances and covariances of moisture and heat content using a multivariate PDF: A study with CLUBB (tag MVCS)

Brian M. Griffin and Vincent E. Larson

University of Wisconsin – Milwaukee, Department of Mathematical Sciences, Milwaukee, WI, USA

*Correspondence to:* Brian M. Griffin (bmg2@uwm.edu)

**Abstract.** Microphysical processes, such as the formation, growth, and evaporation of precipitation, interact with variability and covariances (e.g. fluxes) in moisture and heat content. For instance, evaporation of rain may produce cold pools, which in turn may trigger fresh convection and precipitation. These effects are usually omitted or else crudely parameterized at subgrid scales in weather and climate models.

A more formal approach is pursued here, based on predictive, horizontally averaged equations for the variances, covariances, and fluxes of moisture and heat content. These higher-order moment equations contain microphysical source terms. The microphysics terms can be integrated analytically, given a suitably simple warm-rain microphysics scheme and an approximate assumption about the multivariate distribution of cloud-related and precipitation-related variables. Performing the integrations provides exact expressions within an idealized context.

A large-eddy simulation (LES) of a shallow precipitating cumulus case is performed here, and it indicates that the microphysical effects on (co)variances and fluxes can be large. In some budgets and altitude ranges, they are dominant terms. The analytic expressions for the integrals are implemented in a single-column, higher-order closure model. Interactive single-column simulations agree qualitatively with the LES. The analytic integrations form a parameterization of microphysical effects in their own right, and they also serve as benchmark solutions that can be compared to non-analytic integration methods.

## 1 Introduction

The structure, development, and dissipation of precipitating cumulus clouds are influenced by interactions between microphysical, thermodynamic, and turbulent processes. For example, consider the diurnal cycle of precipitation over land in the tropics. Over tropical land masses, there is a gradual transition from shallow convection in the morning to deep convection several hours later (e.g., Grabowski et al., 2006). Early clouds remain shallow because they entrain dry environmental air (Derbyshire et al., 2004). Successive clouds moisten the environment. The transition to deep convection is aided by a positive feedback involving rain, evaporative cooling, the formation of cold pools, and the triggering of fresh convection and rain. Namely, when precipitation initiates, rain falls and evaporates below cloud base, generating cold pools in the boundary layer. The cold pools, in turn, can lift boundary layer air upwards and thereby trigger new convection (e.g., Kuang and Bretherton, 2006; Khairoutdinov and Randall, 2006; Mapes and Neale, 2011; Böing et al., 2012; Gentine et al., 2016). There may also oc-

cur a negative feedback between thermodynamic variability in clouds and precipitation. Namely, updrafts and turbulent mixing may generate variability in cloud water mixing ratio. Then rain forms preferentially in the moistest part of the cloud, reducing peak cloud water contents, and thereby diminishing variability in cloud water (Khairoutdinov and Randall, 2002). Although these effects may be most pronounced in deep cumuli, which precipitate strongly, they are also present in precipitating shallow

cumulus. Relatedly, some shallow cumulus layers produce cold pools (e.g., Zuidema et al., 2012) and/or exhibit mesoscale organization (e.g., Rauber et al., 2007; Xue et al., 2008).

Some effects of microphysics influence the spatial arrangement of cloud parcels. For instance, precipitation may lead to an increase in cloud diameter or to the development of mesoscale cloud organization (e.g., Kuang and Bretherton, 2006; Khairoutdinov and Randall, 2006; Schlemmer and Hohenegger, 2014). Such effects of microphysics on cloud *structure* will

not be discussed here. Instead, the focus will be on the effects of microphysics on the *variances and covariances* of cloud-related fields. Microphysics affects more than just horizontal averages; it also affects variability. For instance, rain production in the moistest parts of a cloud tends to diminish variability in cloud water. Also, evaporative cooling of rain in cold downdrafts below cloud base may increase the variability in temperature in the subcloud layer. Even though the effects of microphysics on cloud *structure* may be difficult to quantify, the effects of microphysics on *variances and covariances* are simpler to define and

calculate. Those effects appear as well-defined covariance terms on the right-hand side of spatially filtered equations for the scalar variances and turbulent fluxes. These filtered moment equations can be derived rigorously from the governing equations, and the microphysical terms emerge naturally from the derivation. However, most coarse-resolution climate or weather models either treat such effects phenomenologically (e.g., Grandpeix and Lafore, 2010; Rio et al., 2013; Bechtold et al., 2014) or else ignore them entirely.

The microphysical terms in the predictive variance and covariance equations can be parameterized by integrating microphysical formulas over the Probability Density Function (PDF) representing subgrid variability. A primary purpose of this paper is to perform those integrals analytically and to implement the resulting formulas in a particular PDF parameterization, the Cloud Layers Unified By Binormals (CLUBB) model. The needed integrals are set up in Appendix A and are solved by the expressions given in the Supplement. The integrals can be performed analytically because the microphysical formulas that are

integrated are simple power laws (Khairoutdinov and Kogan, 2000), and because it is assumed that the variables involved are distributed according to a multivariate PDF based on normal and lognormal functions (Griffin and Larson, 2016). The analytic solutions to the integrals are used directly as a parameterization. Alternatively, the implementation of the integrals may also serve as a benchmark calculation that is based on idealized (Khairoutdinov-Kogan) microphysics. The benchmark calculation can be used to assess the accuracy and convergence of more general integration methods, as done in Larson and Schanen

(2013). Full evaluation of the use of the integrals as a parameterization is deferred to future work, but for illustrative purposes, single-column CLUBB simulations of a shallow convective case, Rain in Cumulus over the Ocean (RICO) (van Zanten et al., 2011), and a marine stratocumulus case are presented. Budgets from a large-eddy simulation (LES) model are also presented. The LES indicates which variances and covariances are most influenced by microphysical processes. In addition, the LES provides a benchmark budget of each covariance. Each budget term from LES corresponds to a budget term in CLUBB, al-

lowing for a close, term-by-term comparison of model processes. This sort of detailed comparison is infeasible with more phenomenological parameterizations.

To clarify, we note that the microphysical terms we study here appear in the variance and covariance equations, not the grid mean equations. Microphysical effects on the grid means have been studied in several prior works (e.g., Zhang et al., 2002; Larson and Griffin, 2006; Morrison and Gettelman, 2008; Cheng and Xu, 2009; Larson and Griffin, 2013; Griffin and Larson, 2013; Boutle et al., 2014). The microphysical effects on the grid means can shift the subgrid PDF to smaller or larger values, but, unlike the covariance terms, they cannot directly change the shape of the PDF. The microphysical covariance terms are important because 1) they damp variability (i.e. narrow the PDF) via the effects of precipitation rather than turbulence; and 2) they generate variability (i.e. widen the PDF) below cloud via the effects of rain evaporation.

The remainder of the paper is organized as follows. Section 2 overviews the origin of the microphysical terms from the predictive equations, summarizes the microphysics scheme involved in the development of this parameterization, and summarizes the multivariate PDF used by CLUBB. Section 3 describes the test case simulation, the LES used for comparison, and the setup of the CLUBB model. Section 4 compares the budget terms for relevant variances and covariances between the LES and CLUBB. Section 5 contains concluding remarks.

## 2  Mathematical and physical overview

This section indicates where the microphysical terms enter CLUBB's equation set. Microphysical terms have appeared in versions of CLUBB's grid-mean equations for some years, but now microphysics terms also appear in CLUBB's prognostic equations for scalar variances and covariances.

CLUBB is a single-column model (SCM) that predicts variances and covariances involving vertical velocity, moisture, and temperature fields using spatially-filtered moment equations (Golaz et al., 2002; Larson and Golaz, 2005; Larson and Griffin, 2013; Griffin and Larson, 2013). CLUBB uses a multivariate Probability Density Function (PDF) to represent subgrid variability in vertical velocity, moisture, temperature, and hydrometeor fields. The subgrid PDF is used to close the higher-order moment terms found in the predictive moment equations and also to provide information on cloud water and cloud fraction.

CLUBB's PDF and corresponding predictive equation set are based on vertical velocity, $w$, total water mixing ratio, $r_t$, and liquid water potential temperature, $\theta_l$. Total water mixing ratio is defined such that $r_t = r_v + r_c$, where $r_v$ is water vapor mixing ratio and $r_c$ is (liquid) cloud water mixing ratio. Liquid water potential temperature is defined by the equation

$$\theta_l = T_l \left( \frac{p}{p_0} \right)^{-\frac{R_d}{C_{pd}}},$$
(1)

where $p$ is pressure, $p_0$ is a reference pressure of $1.0 \times 10^5$ Pa, $R_d$ is the gas constant for dry air, and $C_{pd}$ is the specific heat of dry air at a constant pressure. Liquid water temperature, $T_l$, is defined as

$$T_l = T - \frac{L_v}{C_{pd}} r_c,$$
(2)

where $T$ is temperature and $L_v$ is the latent heat of vaporization. In subsaturated air, $r_t$ reduces to $r_v$ and $\theta_l$ reduces to potential temperature, $\theta$.

The CLUBB model uses $r_t$ and $\theta_l$ because those variables are conserved with regard to adiabatic processes and phase changes between water vapor and liquid cloud water. However, $r_t$ and $\theta_l$ are not conserved with respect to transfers between precipitation and water vapor or cloud water. As a result, the time-tendency equations for each of $r_t$ and $\theta_l$ include a microphysics tendency term. Omitting all other terms, such as advection, these equations can be written as

$$\frac{\partial r_t}{\partial t} = \ldots + \left.\frac{\partial r_t}{\partial t}\right|_{\mathrm{mc}}, \quad \text{and} \tag{3}$$

$$\frac{\partial \theta_l}{\partial t} = \ldots + \left.\frac{\partial \theta_l}{\partial t}\right|_{\mathrm{mc}}; \tag{4}$$

where $t$ is time, and where $\left.\dfrac{\partial r_t}{\partial t}\right|_{\mathrm{mc}}$ and $\left.\dfrac{\partial \theta_l}{\partial t}\right|_{\mathrm{mc}}$ are the microphysics tendency terms for $r_t$ and $\theta_l$, respectively. They are the source or sink of $r_t$ and $\theta_l$ due to microphysics process rates.

The time-tendency equations are split into mean and turbulent components. For the remainder of this paper, an overbar will denote a mean value, while the prime symbol ($'$) will donate a deviation from the mean value (turbulent value). The Reynolds-averaged predictive equations for grid-box mean fields $\overline{r_t}$ and $\overline{\theta_l}$ include the terms

$$\frac{\partial \overline{r_t}}{\partial t} = \ldots + \overline{\left.\frac{\partial r_t}{\partial t}\right|_{\mathrm{mc}}}, \quad \text{and} \tag{5}$$

$$\frac{\partial \overline{\theta_l}}{\partial t} = \ldots + \overline{\left.\frac{\partial \theta_l}{\partial t}\right|_{\mathrm{mc}}}. \tag{6}$$

The omitted terms in the predictive equations for $\overline{r_t}$ and $\overline{\theta_l}$ are listed in Golaz et al. (2002), with the only change being that the CLUBB equation set is now written in anelastic form.

In order to obtain the fields necessary to generate the PDF, CLUBB also contains predictive equations for the subgrid variances and covariances involving $w$, $r_t$, and $\theta_l$. The fields that contain a microphysics term are $\overline{w'r_t'}$, $\overline{w'\theta_l'}$, $\overline{r_t'^2}$, $\overline{\theta_l'^2}$, and $\overline{r_t'\theta_l'}$. The Reynolds-averaged predictive equations for these subgrid variances and covariances include the terms:

$$\frac{\partial \overline{w'r_t'}}{\partial t} = \ldots + \overline{w'\left.\frac{\partial r_t}{\partial t}\right|'_{\mathrm{mc}}}, \tag{7}$$

$$\frac{\partial \overline{w'\theta_l'}}{\partial t} = \ldots + \overline{w'\left.\frac{\partial \theta_l}{\partial t}\right|'_{\mathrm{mc}}}, \tag{8}$$

$$\frac{\partial \overline{r_t'^2}}{\partial t} = \ldots + 2\,\overline{r_t'\left.\frac{\partial r_t}{\partial t}\right|'_{\mathrm{mc}}}, \tag{9}$$

$$\frac{\partial \overline{\theta_l'^2}}{\partial t} = \ldots + 2\,\overline{\theta_l'\left.\frac{\partial \theta_l}{\partial t}\right|'_{\mathrm{mc}}}, \quad \text{and} \tag{10}$$

$$\frac{\partial \overline{r_t'\theta_l'}}{\partial t} = \ldots + \overline{r_t'\left.\frac{\partial \theta_l}{\partial t}\right|'_{\mathrm{mc}}} + \overline{\theta_l'\left.\frac{\partial r_t}{\partial t}\right|'_{\mathrm{mc}}}. \tag{11}$$

The full forms, including all omitted terms, of the predictive equations for $\overline{w'r_t'}$, $\overline{w'\theta_l'}$, $\overline{r_t'^2}$, $\overline{\theta_l'^2}$, and $\overline{r_t'\theta_l'}$ are given by Eq. (26), Eq. (27), Eq. (28), Eq. (29), and Eq. (30), respectively, in Section 4.

If $r_t$ and $\theta_l$ were extended to include precipitation, the extended variables would be conserved with respect to transfers between hydrometeors and water vapor or cloud water (microphysics process rates). This is a simplification, but we do not choose to extend the variables to include precipitation because it would lead to a complication, namely, it would cause the microphysical effects to appear in sedimentation terms, and the sedimentation terms contain vertical derivatives, unlike the process rate terms. Furthermore, some turbulent components of the sedimentation term contain a vertical derivative within a horizontal average. To illustrate, consider a hydrometeor-inclusive total water mixing ratio, denoted $r_T$, such that $r_T = r_v + r_c + r_r$, where $r_r$ is rain water mixing ratio. For simplicity, $r_r$ will be the only hydrometeor considered in the microphysics. The microphysics term on the right-hand side of the Reynolds-averaged $\overline{r_T'^2}$ predictive equation would have the form

$$2\,\overline{r_T'\left(-\frac{1}{\rho_s}\frac{\partial \rho_s V_{r_r} r_r}{\partial z}\right)'} = -\frac{2}{\rho_s}\,\overline{r_T'\frac{\partial \rho_s \overline{V_{r_r}}\,\overline{r_r'}}{\partial z}} - \frac{2}{\rho_s}\,\overline{r_T'\frac{\partial \rho_s V_{r_r}'\,\overline{r_r}}{\partial z}} - \frac{2}{\rho_s}\,\overline{r_T'\frac{\partial \rho_s V_{r_r}'\,r_r'}{\partial z}},$$

where $V_{r_r}$ is the sedimentation velocity of $r_r$, $\rho_s$ is the dry, base-state air density, and $z$ is height. Every predictive moisture or temperature (co)variance equation would contain terms analogous to the above sedimentation terms. Since these terms contain vertical derivatives ($\partial/\partial z$) embedded within integrals over the horizontal, they are difficult to treat analytically and cannot be described solely by a multivariate subgrid PDF at a single vertical grid level. For this reason, CLUBB's calculations of the microphysics terms use $r_t$ and $\theta_l$ defined in terms of cloud water, not precipitation.

## 2.1 KK microphysics

The preceding section describes where microphysical tendencies, of any kind, enter CLUBB's equation set. This section describes how the microphysical tendencies are related to the specific processes of autoconversion, accretion, and evaporation.

The source terms for the model predictive equations require microphysical process rates from a microphysics scheme. The scheme used here is the warm microphysics scheme described in Khairoutdinov and Kogan (2000, hereafter KK). KK is a two-moment scheme that predicts $r_r$ and rain drop concentration (per unit mass), $N_r$. It was developed by using the least squares method to find a "best-fit" curve through microphysical rate data that was generated by simulating a drizzling stratocumulus case using an explicit (or "bin") microphysics scheme.

The KK scheme was chosen because of its simplicity. It expresses microphysical rates as power laws of two or three variables, which means that the product of a microphysical rate and the corresponding PDF is always integrable. More recently, the coefficients and exponents in the KK scheme have been tailored to cumulus clouds (Kogan, 2013). The Kogan scheme is covered by the analytic integrals presented in this paper because they are generalized for arbitrary coefficients and exponents. However, this paper uses the original KK coefficients and exponents because the KK scheme has been widely used for a variety of cloud types and is adequate for our idealized purposes.

The KK warm microphysics scheme produces $r_r$ through the processes of autoconversion (collision) and accretion (collection). These processes produce rain water, deplete cloud water, and leave water vapor unchanged. As a result, these processes increase the value of $\theta_l$, as shown by Eq. (1) and Eq. (2), and decrease the value of $r_t$. Evaporation reduces $r_r$ as rain falls

through subsaturated air. Condensational growth does not apply to rain water in CLUBB. Instead, all supersaturation is automatically applied to cloud water. When rain water evaporates, cloud water remains unchanged, and $r_t$ increases due to the increase in water vapor. Meanwhile, evaporative cooling decreases $\theta_l$ due to the decrease in temperature.

The relationship of all three KK microphysics tendencies to the $r_t$ microphysics tendency can be written as

$$\left.\frac{\partial r_t}{\partial t}\right|_{mc} = -\left.\frac{\partial r_r}{\partial t}\right|_{auto} - \left.\frac{\partial r_r}{\partial t}\right|_{accr} - \left.\frac{\partial r_r}{\partial t}\right|_{evap}, \tag{12}$$

where $\left.\frac{\partial r_r}{\partial t}\right|_{auto}$ is the rate of change of $r_r$ due to the process of autoconversion, $\left.\frac{\partial r_r}{\partial t}\right|_{accr}$ is the rate of change of $r_r$ due to the process of accretion, and $\left.\frac{\partial r_r}{\partial t}\right|_{evap}$ is the rate of change of $r_r$ due to the process of evaporation. Note that when evaporation occurs, $\left.\frac{\partial r_r}{\partial t}\right|_{evap} < 0$. The relationship of all three tendencies to $\theta_l$ microphysics tendency can be written as

$$\left.\frac{\partial \theta_l}{\partial t}\right|_{mc} = \frac{L_v}{C_{pd}}\left(\frac{p}{p_0}\right)^{-\frac{R_d}{C_{pd}}}\left(\left.\frac{\partial r_r}{\partial t}\right|_{auto} + \left.\frac{\partial r_r}{\partial t}\right|_{accr} + \left.\frac{\partial r_r}{\partial t}\right|_{evap}\right). \tag{13}$$

The decrease in temperature from the evaporation of a unit of rain water is the same as the decrease in temperature from the evaporation of the same amount of cloud water.

The Reynolds-averaged microphysics term in the predictive equation for $\overline{w'r_t'}$, as found in Eq. (7), is rewritten as

$$\overline{w'\left.\frac{\partial r_t}{\partial t}\right|_{mc}'} = -\overline{w'\left.\frac{\partial r_r}{\partial t}\right|_{auto}'} - \overline{w'\left.\frac{\partial r_r}{\partial t}\right|_{accr}'} - \overline{w'\left.\frac{\partial r_r}{\partial t}\right|_{evap}'}. \tag{14}$$

Likewise, the Reynolds-averaged microphysics term in the predictive equation for $\overline{w'\theta_l'}$, as found in Eq. (8), is rewritten as

$$\overline{w'\left.\frac{\partial \theta_l}{\partial t}\right|_{mc}'} = \frac{L_v}{C_{pd}}\left(\frac{\bar{p}}{p_0}\right)^{-\frac{R_d}{C_{pd}}}\left(\overline{w'\left.\frac{\partial r_r}{\partial t}\right|_{auto}'} + \overline{w'\left.\frac{\partial r_r}{\partial t}\right|_{accr}'} + \overline{w'\left.\frac{\partial r_r}{\partial t}\right|_{evap}'}\right). \tag{15}$$

Any variability of $p$ within the grid box is ignored for simplicity. Additionally, the $-R_d/C_{pd}$ exponent would greatly limit the effects of variability of $p$ on the solution. As a result, $\bar{p}$ is used in the equation. In the predictive equation for $\overline{r_t'^2}$, Eq. (9), the microphysics term becomes

$$\overline{r_t'\left.\frac{\partial r_t}{\partial t}\right|_{mc}'} = -\overline{r_t'\left.\frac{\partial r_r}{\partial t}\right|_{auto}'} - \overline{r_t'\left.\frac{\partial r_r}{\partial t}\right|_{accr}'} - \overline{r_t'\left.\frac{\partial r_r}{\partial t}\right|_{evap}'}. \tag{16}$$

In the predictive equation for $\overline{\theta_l'^2}$, Eq. (10), the microphysics term becomes

$$\overline{\theta_l'\left.\frac{\partial \theta_l}{\partial t}\right|_{mc}'} = \frac{L_v}{C_{pd}}\left(\frac{\bar{p}}{p_0}\right)^{-\frac{R_d}{C_{pd}}}\left(\overline{\theta_l'\left.\frac{\partial r_r}{\partial t}\right|_{auto}'} + \overline{\theta_l'\left.\frac{\partial r_r}{\partial t}\right|_{accr}'} + \overline{\theta_l'\left.\frac{\partial r_r}{\partial t}\right|_{evap}'}\right). \tag{17}$$

The Reynolds-averaged microphysics terms in the predictive equation for $\overline{r_t'\theta_l'}$, as found in Eq. (11), are rewritten as

$$\overline{\theta_l'\left.\frac{\partial r_t}{\partial t}\right|_{mc}'} = -\overline{\theta_l'\left.\frac{\partial r_r}{\partial t}\right|_{auto}'} - \overline{\theta_l'\left.\frac{\partial r_r}{\partial t}\right|_{accr}'} - \overline{\theta_l'\left.\frac{\partial r_r}{\partial t}\right|_{evap}'}, \text{ and} \tag{18}$$

$$\overline{r_t' \left.\frac{\partial \theta_l}{\partial t}\right|_{mc}}' = \frac{L_v}{C_{pd}} \left(\frac{\bar{p}}{p_0}\right)^{-\frac{R_d}{C_{pd}}} \left( \overline{r_t' \left.\frac{\partial r_r}{\partial t}\right|_{auto}}' + \overline{r_t' \left.\frac{\partial r_r}{\partial t}\right|_{accr}}' + \overline{r_t' \left.\frac{\partial r_r}{\partial t}\right|_{evap}}' \right). \tag{19}$$

The above equation set contains nine individual microphysical covariance terms, each involving one of $w$, $r_t$, or $\theta_l$ with one of autoconversion, accretion, or evaporation rate. These terms can be parameterized through use of the PDF method.

## 2.2 PDF method

The multivariate PDF used by CLUBB consists of $w$, $r_t$, $\theta_l$, all hydrometeor species used by the selected microphysics scheme (in the case of KK microphysics, $r_r$ and $N_r$), and an extended cloud droplet concentration, $N_{cn}$, which is equal to cloud droplet concentration, $N_c$, within cloud, but has a positive value outside of cloud (Griffin and Larson, 2016). CLUBB's PDF is a weighted mixture, or sum, of two multivariate normal/lognormal functions. Each multivariate function is known as a PDF component.

When variables are integrated out of the multivariate PDF, a marginal PDF consisting of fewer variables remains. When all variables but one are integrated out of the PDF, the result is a univariate marginal or individual marginal. The individual marginal for each of $w$, $r_t$, and $\theta_l$ is a two-component normal (also known as a binormal) distribution. The two-component shape allows skewness to be included in model fields. The individual marginal for $N_{cn}$ is assumed to be a (single) lognormal distribution.

The individual marginal for each of $r_r$ and $N_r$ is delta-lognormal *within each PDF component* (Griffin and Larson, 2016). Each PDF component can contain precipitating and precipitation-less regions. The fraction of each PDF component that contains any hydrometeor species (other than cloud liquid water) is known as the component's precipitation fraction. The precipitation-less region is represented by a delta at 0 for all hydrometeor species. Within precipitation, a lognormal distribution is used to represent a hydrometeor species. The lognormal distributions can differ between the two components, so that when the components are summed to form the overall distribution, a delta double lognormal (DDL) distribution results.

The PDF method for parameterizing the nine microphysics covariance terms requires analytic integration over the multivariate PDF. As listed in Griffin and Larson (2016), the general form of a multivariate PDF of $n$ components and $D$ variables, where $D$ can be all the variables involved in the PDF or any subset of those, is given by

$$P(x_1, x_2, \ldots, x_D) = \sum_{i=1}^{n} \xi_{(i)} P_{(i)}(x_1, x_2, \ldots, x_D), \tag{20}$$

where $\xi_{(i)}$ is the mixture fraction, or relative weight of the $i$th PDF component. The sum of the mixture fractions is equal to 1.

The $D$ variables listed are categorized, and the first $J$ variables are normally distributed in each PDF component ($w$, $r_t$, and $\theta_l$), the next $K$ variables are lognormally distributed ($N_{cn}$), and the last $\Omega$ variables are the hydrometeor species that are distributed delta-lognormally in each PDF component ($r_r$ and/or $N_r$). The equation for the $i$th PDF component is

$$\begin{aligned} P_{(i)}(x_1, x_2, \ldots, x_D) = {} & f_{p(i)} P_{(J,K+\Omega)(i)}(x_1, x_2, \ldots, x_D) \\ & + \left(1 - f_{p(i)}\right) P_{(J,K)(i)}(x_1, x_2, \ldots, x_{J+K}) \left( \prod_{\epsilon=J+K+1}^{D} \delta(x_\epsilon) \right), \end{aligned} \tag{21}$$

where $f_{p(i)}$ is the precipitation fraction in the $i$th PDF component. The subscripts in the $i$th component, $P_{(J,K)(i)}$ or $P_{(J,K+\Omega)(i)}$, denote the number of normal variates, $J$, and the number of lognormal variates, $K$ or $K + \Omega$, used in Eq. (22).

Both the precipitating and precipitation-less portions (sub-components) of Eq. (21) contain a hybrid normal/lognormal distribution of $m$ variables, where the first $j$ variables are normally distributed and the remaining $k$ variables are lognormally

distributed. The general form of this multivariate normal/lognormal PDF is given by (Fletcher and Zupanski, 2006)

$$
P_{(j,k)(i)}(x_1, x_2, \ldots, x_m) = \frac{1}{(2\pi)^{\frac{m}{2}} |\mathbf{\Sigma}_{(i)}|^{\frac{1}{2}}} \left( \prod_{\tau=j+1}^{m} \frac{1}{x_\tau} \right)
$$
$$
\times \exp \left\{ -\frac{1}{2} \left( \boldsymbol{x} - \boldsymbol{\mu}_{(i)} \right)^{\mathrm{T}} \mathbf{\Sigma}_{(i)}^{-1} \left( \boldsymbol{x} - \boldsymbol{\mu}_{(i)} \right) \right\}, \tag{22}
$$

where $\boldsymbol{x}$ is a $m \times 1$ vector of the variables (in normal-space) in the PDF and $\boldsymbol{\mu}_{(i)}$ is a $m \times 1$ vector of the (normal-space) PDF sub-component means. The transpose of the vector is denoted T. The $m \times m$ (normal-space) covariance matrix is denoted $\mathbf{\Sigma}_{(i)}$ and its determinant is denoted $|\mathbf{\Sigma}_{(i)}|$ (Fletcher and Zupanski, 2006).

Eq. (22) lists the general functional form for the subgrid PDF, but specific examples of marginals for a single mixture component are written out in the Supplement to this article. These examples help provide intuition about the shape of the PDF. For instance, a univariate normal marginal of the PDF is written in Eq. (S7), and a univariate lognormal is written in Eq. (S8). A normal distribution is symmetric, extends from $(-\infty, +\infty)$, and has short tails. A lognormal distribution, on the other hand, has a skewed shape that is useful for representing the distribution of a quantity such as rain mixing ratio. Such distributions

are non-negative and often have a peak at low values and a long tail of larger values extending to the right. They are not well represented by normal distributions.

Also useful for gaining intuition are the bivariate marginals listed in Section S3 of the Supplement. A normal-normal bivariate form is listed in Eq. (S4), a lognormal-lognormal form is listed in Eq. (S6), and a hybrid normal-lognormal form is listed in Eq. (S5). Where a lognormal variate appears, the corresponding axis takes on only non-negative values and has a long tail.

Which bivariate form is used depends on which functional forms are used to represent the variates of interest, e.g., rain mixing ratio (lognormal) or extended cloud water mixing ratio (normal).

Using a two-component PDF requires a method to divide one *overall* (grid-box) mean value of a variable into two PDF *component* mean values of that variable. Likewise, one overall variance needs to be split into two PDF component standard deviations. The multivariate PDF also requires information on the correlations between variables.

The PDF component means, standard deviations, and correlations involving $w$, $r_t$, and $\theta_l$, as well as the mixture fractions, are calculated according to the Analytic Double Gaussian 1 (ADG1) PDF presented in Section (d) of the Appendix of Larson et al. (2002). The overall (grid-box) precipitation fraction is set to the maximum cloud fraction found at or above that grid level (Morrison and Gettelman, 2008). The calculation of the component precipitation fractions $f_{p(i)}$ from the overall precipitation fraction are outlined in Griffin and Larson (2016). Also described there is the calculation of the PDF component means and

standard deviations involving $N_{cn}$, $r_r$, and $N_r$. Interactive CLUBB runs prescribe a constant ratio of the in-precipitation variance to the square of the in-precipitation mean for $r_r$ and $N_r$. Additionally, all remaining correlations between variables are prescribed constants.

The covariance of PDF variables $x_1$ and $x_2$ can be calculated by

$$\overline{x_1' x_2'} = \int \int (x_1 - \overline{x_1})(x_2 - \overline{x_2}) P(x_1, x_2) \, dx_2 \, dx_1. \tag{23}$$

The covariance of a PDF variable and a microphysics function (written in terms of PDF variables) can be calculated in the same manner. For example, the covariance of $\theta_l$ and KK evaporation rate found in Eq. (17) and Eq. (18) can be rewritten as

$$\overline{\theta_l' \left. \frac{\partial r_r}{\partial t}\right|_{\mathrm{evap}}'} = \overline{\left(\theta_l - \overline{\theta_l}\right) \left( \left.\frac{\partial r_r}{\partial t}\right|_{\mathrm{evap}} - \overline{\left.\frac{\partial r_r}{\partial t}\right|_{\mathrm{evap}}} \right)}, \tag{24}$$

where mean evaporation rate, $\overline{\left.\dfrac{\partial r_r}{\partial t}\right|_{\mathrm{evap}}}$, is also calculated by integrating over the PDF (Supplement to Griffin and Larson, 2016; Larson and Griffin, 2013). The KK evaporation rate can be written as a function of $\theta_l$, $r_t$, $r_r$, and $N_r$, so here it will be referred to as $\mathrm{EV}(\theta_l, r_t, r_r, N_r)$. The covariance of $\theta_l$ and KK evaporation rate is calculated by

$$\overline{\theta_l' \left. \frac{\partial r_r}{\partial t}\right|_{\mathrm{evap}}'} = \int \int \int \int \left(\theta_l - \overline{\theta_l}\right) \left( \mathrm{EV}(\theta_l, r_t, r_r, N_r) - \overline{\mathrm{EV}(\theta_l, r_t, r_r, N_r)} \right)$$
$$\times P(\theta_l, r_t, r_r, N_r) \, dN_r \, dr_r \, dr_t \, d\theta_l. \tag{25}$$

The remaining eight covariances involving microphysical functions are calculated in the same manner. Further and more detailed description of this method can be found in Appendix A.

## 3   Test case and model setups

To perform an initial test of the parameterization, we choose the Rain in Cumulus over the Ocean (RICO) model intercomparison case of a precipitating shallow cumulus layer (van Zanten et al., 2011). The intercomparison model configuration is based on a field study conducted off the coast of Antigua and Barbuda (Rauber et al., 2007). RICO uses prescribed radiative and large-scale forcings for temperature and moisture, as well as prescribed large-scale subsidence. These quantities vary with altitude but are constant over time. The surface fluxes are calculated using bulk aerodynamic equations. The simulation was run for a period of 72 hours.

RICO was chosen as a test case for two main reasons. First, ice microphysics is not necessary for a shallow trade-wind cumulus case; hence, a warm microphysics scheme is sufficient. Secondly, RICO is a partly cloudy case that precipitates over a small portion of the horizontal domain and contains significant variance of $r_r$ within the precipitating region. These factors lead to significant microphysical effects on the subgrid variances and covariances.

In order to demonstrate that the effects of microphysics on the same subgrid variances and covariances are negligible in a stratocumulus test case, we also ran the drizzling stratocumulus test case based on research flight two (RF02) of the second Dynamics and Chemistry of Marine Stratocumulus (DYCOMS-II) field study (Ackerman et al., 2009; Wyant et al., 2007). DYCOMS-II RF02 uses prescribed large-scale subsidence and constant surface fluxes. Radiative heating is calculated as described in Ackerman et al. (2009). The simulation was run for a period of six hours.

In order to provide benchmarks for comparison, large-eddy simulations (LESs) of RICO and DYCOMS-II RF02 were run using the System for Atmospheric Modeling (SAM) (Khairoutdinov and Randall, 2003). SAM uses an anelastic equation set that predicts all three components of velocity, total water mixing ratio, liquid water static energy, and hydrometeor fields (based on the selected microphysics scheme). A third-order Adams-Bashforth time-stepping scheme is used to advance the predictive equations of motion. The predictive fields are advected by the second-order MPDATA (multidimensional positive definite advection transport algorithm) scheme (Smolarkiewicz and Grabowski, 1990). The subgrid-scale fluxes are computed by a 1.5-order subgrid-scale turbulence kinetic energy closure.

The SAM LES of RICO was run using KK microphysics. SAM's implementation of KK microphysics predicts both $r_r$ and $N_r$. Cloud water mixing ratio, $r_c$, is calculated using a simple saturation adjustment scheme. Cloud droplet concentration, $N_c$, is set to a constant value of $70 \text{ cm}^{-3}$ within cloud. SAM uses a fixed, Cartesian grid. For the RICO case, a $256 \times 256$ horizontal grid is used with a grid spacing of 100 m in each direction. The vertical grid contains 100 levels with 40 m grid spacing, spanning a domain of depth 4000 m. The model time step is 1 s, and horizontally averaged statistical profiles are sampled and output every 60 s. SAM uses periodic boundary conditions at the lateral boundaries and a rigid lid at the top of the domain.

The single-column CLUBB simulation of RICO was run using the analytically upscaled version of KK microphysics, including the microphysical effects on the predictive variances and covariances as described in Section 2. In addition to $\overline{r_t}$, $\overline{\theta_l}$, $\overline{w'r_t'}$, $\overline{w'\theta_l'}$, $\overline{r_t'^2}$, $\overline{\theta_l'^2}$, and $\overline{r_t'\theta_l'}$, CLUBB also predicts the variance and third-order central moment of vertical velocity ($\overline{w'^2}$ and $\overline{w'^3}$, respectively), the mean and variance of the horizontal west-east wind component ($\overline{u}$ and $\overline{u'^2}$, respectively), the mean and variance of the horizontal south-north wind component ($\overline{v}$ and $\overline{v'^2}$, respectively), and the mean of each hydrometeor field involved in the microphysics ($\overline{r_r}$ and $\overline{N_r}$ for KK microphysics). The anelastic approximation is used in all predictive equations. CLUBB calculates $\overline{r_c}$ by using a simple saturation adjustment and integration over the subgrid PDF. Just as in SAM LES, cloud droplet concentration is set to a constant value in cloud for the RICO case. CLUBB uses a vertically stretched grid containing 37 levels covering a domain of depth 4904 m. The model time step is 180 s, and statistical profiles are sampled and output at every model time step.

The SAM LES of DYCOMS-II RF02 also was run using KK microphysics. Cloud droplet concentration is set to a constant value of $55 \text{ cm}^{-3}$ within cloud. The horizontal resolution is 50 m and 128 grid boxes are used in each horizontal direction. The model uses a vertical grid containing 96 levels and covers a domain of depth 1459 m. The time step is 0.5 s. The single-column CLUBB simulation of DYCOMS-II RF02 is run using the analytically upscaled version of KK microphysics and the same constant cloud droplet concentration used for SAM LES. CLUBB uses a vertically streched grid covering a domain of depth 1600 m and a time step of 60 s.

In the following analysis, profiles of the SAM LES and CLUBB SCM budget terms for the $\overline{w'r_t'}$, $\overline{w'\theta_l'}$, $\overline{r_t'^2}$, $\overline{\theta_l'^2}$, and $\overline{r_t'\theta_l'}$ fields are time-averaged over the last half (36 hours) of the RICO simulation (minutes 2160 through 4320). The RICO fields are in an approximately steady state during this time period. The DYCOMS-II RF02 profiles are time-averaged over the last hour (minutes 300 through 360) of the simulation.

## 4 Results

### 4.1 RICO precipitating cumulus

In order to assess which physical processes are most important, the LES budget terms for turbulent fields are analyzed for the RICO precipitating cumulus case. Additionally, the LES budgets and CLUBB's budgets are compared in order to assess the accuracy of CLUBB's budget terms.

Unlike the LES, the CLUBB budget terms for turbulent fields are taken directly from the predictive equation set. The anelastic predictive equations for the turbulent fluxes $\overline{w'r'_t}$ and $\overline{w'\theta'_l}$ are given by

$$\frac{\partial \overline{w'r'_t}}{\partial t} = \underbrace{-\frac{1}{\rho_s}\frac{\partial \rho_s \overline{w}\,\overline{w'r'_t}}{\partial z} - \frac{1}{\rho_s}\frac{\partial \rho_s \overline{w'^2 r'_t}}{\partial z}}_{\text{advection}} \underbrace{- \overline{w'^2}\frac{\partial \overline{r_t}}{\partial z} - \overline{w'r'_t}\frac{\partial \overline{w}}{\partial z}}_{\text{production}} \underbrace{- \frac{1}{\rho_s}\overline{r'_t \frac{\partial p'}{\partial z}}}_{\text{pressure}}$$

$$\underbrace{+ \frac{g}{\theta_{vs}}\overline{r'_t \theta'_v}}_{\text{buoyancy}} \underbrace{+ \varepsilon_{w\,r_t}}_{\text{diffusion}} \underbrace{+ \overline{w'\frac{\partial r_t}{\partial t}\Big|'_{\text{mc}}}}_{\text{microphysics}}, \qquad \text{and} \tag{26}$$

$$\frac{\partial \overline{w'\theta'_l}}{\partial t} = \underbrace{-\frac{1}{\rho_s}\frac{\partial \rho_s \overline{w}\,\overline{w'\theta'_l}}{\partial z} - \frac{1}{\rho_s}\frac{\partial \rho_s \overline{w'^2 \theta'_l}}{\partial z}}_{\text{advection}} \underbrace{- \overline{w'^2}\frac{\partial \overline{\theta_l}}{\partial z} - \overline{w'\theta'_l}\frac{\partial \overline{w}}{\partial z}}_{\text{production}} \underbrace{- \frac{1}{\rho_s}\overline{\theta'_l \frac{\partial p'}{\partial z}}}_{\text{pressure}}$$

$$\underbrace{+ \frac{g}{\theta_{vs}}\overline{\theta'_l \theta'_v}}_{\text{buoyancy}} \underbrace{+ \varepsilon_{w\,\theta_l}}_{\text{diffusion}} \underbrace{+ \overline{w'\frac{\partial \theta_l}{\partial t}\Big|'_{\text{mc}}}}_{\text{microphysics}}, \tag{27}$$

where $g$ is gravity and $\theta_v$ is virtual potential temperature. The dry, anelastic base-state values of air density, $\rho_s$, and $\theta_v$, denoted $\theta_{vs}$, vary only with altitude. The higher-order turbulent advection terms, $\overline{w'^2 r'_t}$ and $\overline{w'^2 \theta'_l}$, are closed using the PDF (Larson and Golaz, 2005). The pressure terms are parameterized following André et al. (1978) (see also Golaz et al. (2002)). The slow (return-to-isotropy) term is approximated by Newtonian dampling. The buoyancy terms are closed by linearizing and then integrating over the PDF (Larson et al., 2002). The terms denoted $\varepsilon_{w\,r_t}$ and $\varepsilon_{w\,\theta_l}$ are background numerical vertical diffusion terms (Golaz et al., 2002).

As in CLUBB, the SAM LES budgets for the horizontally averaged turbulent fluxes contain advective transport terms and turbulent (gradient) production terms, which both ultimately arise from the 3D advection of $w$, $r_t$, and $\theta_l$. The turbulent production terms generate variability when the vertical derivative of the mean field is non-zero. SAM also records the effects of pressure, buoyancy, and microphysics on the turbulent fluxes. SAM's budget term for diffusion of $\overline{w'r'_t}$ and $\overline{w'\theta'_l}$ records the effects of diffusion associated with the subgrid TKE scheme. In Fig. 1 and Fig. 2, following Khairoutdinov and Randall (2002), the SAM LES budget terms for buoyancy and pressure are combined because they are both large compared to other terms, yet are in close equilibrium because of the quasi-hydrostatic balance of perturbation buoyancy and perturbation pressure gradient. The CLUBB buoyancy and pressure terms have been combined in an analogous manner.

The SAM LES turbulent flux budgets show that the largest terms are pressure+buoyancy, which usually acts as a net sink of turbulent flux, and turbulent production, which acts as a source of turbulent flux (see Fig. 1(a) and Fig. 2(a)). Another major

term in the budget is the (turbulent) advection term. The turbulent advection term (e.g. $-(1/\rho_s)\partial(\rho_s \overline{w'^2 r_t'})/\partial z$) has a mass-weighted vertical integral of zero. That is, averaged in the vertical, it is neither a net source nor a net sink. Instead, it takes the excess variability at some altitudes and transports it to regions with a deficit of variability. The microphysics term is a sink of turbulent flux in the cloudy layer, a layer which spans the altitude range from 500 m to 3000 m. The microphysics term is more

significant for $\overline{w'\theta_l'}$ than for $\overline{w'r_t'}$, but even for $\overline{w'r_t'}$, it is non-negligible.

CLUBB's turbulent flux budgets usually agree qualitatively with those from LES (Fig. 1(b) and Fig. 2(b)). CLUBB's advection terms have approximately the correct shape, although they are usually too small in magnitude. In CLUBB, the buoyancy+pressure and turbulent production terms are dominant, as in SAM LES, but in CLUBB's RICO simulation their magnitudes are larger than in SAM LES.

The microphysics terms in both the $\overline{w'r_t'}$ and $\overline{w'\theta_l'}$ budgets have the same signs and close to the same peak magnitudes as their counterparts in the LES. However, in CLUBB, the range of altitudes where the microphysics budget terms have significant values is shifted lower than in SAM LES. This occurs because $\overline{r_r}$ peaks at a lower altitude in CLUBB than in SAM LES. The lower-altitude peak in rain, in turn, occurs because there is too much evaporation near cloud top, as shown in Fig. 7(a) of Griffin and Larson (2016). As noted there, the excessive evaporation is caused by an excessively long-tailed marginal subgrid

PDF of saturation deficit, which extends to unrealistically dry values. The excessive evaporation near cloud top also causes a similar problem in the microphysical terms in the other budgets presented below. See Griffin and Larson (2016) for more details.

The CLUBB anelastic predictive equations for the scalar variances $\overline{r_t'^2}$ and $\overline{\theta_l'^2}$, and the covariance $\overline{r_t'\theta_l'}$, are given by

$$\frac{\partial \overline{r_t'^2}}{\partial t} = \underbrace{-\frac{1}{\rho_s}\frac{\partial \rho_s \overline{w}\,\overline{r_t'^2}}{\partial z} - \frac{1}{\rho_s}\frac{\partial \rho_s \overline{w'r_t'^2}}{\partial z}}_{\text{advection}} \underbrace{-2\,\overline{w'r_t'}\frac{\partial \overline{r_t}}{\partial z}}_{\text{production}} \underbrace{+\varepsilon_{r_t r_t}}_{\text{diss+diff}} \underbrace{+2\,\overline{r_t'\left.\frac{\partial r_t}{\partial t}\right|'_{\text{mc}}}}_{\text{microphysics}}, \tag{28}$$

$$\frac{\partial \overline{\theta_l'^2}}{\partial t} = \underbrace{-\frac{1}{\rho_s}\frac{\partial \rho_s \overline{w}\,\overline{\theta_l'^2}}{\partial z} - \frac{1}{\rho_s}\frac{\partial \rho_s \overline{w'\theta_l'^2}}{\partial z}}_{\text{advection}} \underbrace{-2\,\overline{w'\theta_l'}\frac{\partial \overline{\theta_l}}{\partial z}}_{\text{production}} \underbrace{+\varepsilon_{\theta_l \theta_l}}_{\text{diss+diff}} \underbrace{+2\,\overline{\theta_l'\left.\frac{\partial \theta_l}{\partial t}\right|'_{\text{mc}}}}_{\text{microphysics}}, \text{ and} \tag{29}$$

$$\frac{\partial \overline{r_t'\theta_l'}}{\partial t} = \underbrace{-\frac{1}{\rho_s}\frac{\partial \rho_s \overline{w}\,\overline{r_t'\theta_l'}}{\partial z} - \frac{1}{\rho_s}\frac{\partial \rho_s \overline{w'r_t'\theta_l'}}{\partial z}}_{\text{advection}} \underbrace{-\overline{w'r_t'}\frac{\partial \overline{\theta_l}}{\partial z} - \overline{w'\theta_l'}\frac{\partial \overline{r_t}}{\partial z}}_{\text{production}} \underbrace{+\varepsilon_{r_t \theta_l}}_{\text{diss+diff}}$$

$$\underbrace{+\overline{r_t'\left.\frac{\partial \theta_l}{\partial t}\right|'_{\text{mc}}} + \overline{\theta_l'\left.\frac{\partial r_t}{\partial t}\right|'_{\text{mc}}}}_{\text{microphysics}}. \tag{30}$$

As in the predictive equations for the fluxes, the higher-order turbulent advection terms, $\overline{w'r_t'^2}$, $\overline{w'\theta_l'^2}$, and $\overline{w'r_t'\theta_l'}$, are closed

using the PDF (Larson and Golaz, 2005). The terms denoted $\varepsilon_{r_t r_t}$, $\varepsilon_{\theta_l \theta_l}$, and $\varepsilon_{r_t \theta_l}$ each contain a dissipation term (parameterized in CLUBB as Newtonian damping) that reduces the magnitude of the turbulent field, as well as a background numerical vertical diffusion term (Golaz et al., 2002; André et al., 1978).

The SAM LES budgets for the horizontally averaged turbulent (co)variances contain advective transport terms and turbulent (gradient) production terms, as well as microphysics terms. In Fig. 3, Fig. 4, and Fig. 5, the diffusion and dissipation terms are combined for both SAM and CLUBB. Both SAM and CLUBB contain vertical diffusion, with SAM's associated with TKE. However, SAM's subgrid TKE is also used to diffuse fields horizontally. Horizontal diffusion smooths out a model field across the grid level, reducing the variances and covariances of model fields. In CLUBB, this effect is parameterized by the dissipation (Newtonian damping) term.

The SAM LES budgets for $\overline{r_t'^2}$, $\overline{\theta_l'^2}$, and $\overline{r_t'\theta_l'}$ show that microphysics is a dominant term in the upper half of the cloud layer. At those levels, microphysics is balanced by turbulent production and turbulent advection (at higher altitudes) (Figs. 3(a), 4(a), and 5(a)). Near cloud base, the budget is predominantly a balance of advection and production. The dissipation/diffusion terms are smaller, but not negligible.

The time-averaged CLUBB SCM budgets found in Fig. 3(b), Fig. 4(b), and Fig. 5(b) show that the CLUBB scalar (co)variance budgets are qualitatively similar to the LES budgets. The microphysics term in the $\overline{r_t'^2}$ budget has the correct sign and is a significant sink term, and the shape of the profile of the advection and production terms qualitatively resemble the LES. CLUBB's dissipation term is too large, but the microphysics terms in the $\overline{\theta_l'^2}$ and $\overline{r_t'\theta_l'}$ budgets are dominant terms in the cloudy layer, just as in the LES. The production terms largely balance the microphysics terms. The advection terms are too small in magnitude relative to the other terms, but have approximately the right shape.

The figures show that the microphysics terms are sink terms in the cloudy layer, reducing the variances and the magnitudes of the covariances, for all five of these turbulent fields. Physically, this happens because cumulus clouds arise in the regions of the horizontal domain that are moister than average. Additionally, cloudy regions are usually associated with updrafts (where vertical velocity is greater than average) in a cumulus regime. Within cloud, the moistest regions contain the greatest amount of cloud (liquid) water. The microphysics processes of autoconversion and accretion occur only in cloud and at greater rates in regions with a greater amount of cloud water. When autoconversion and accretion occur, rain water is produced at the expense of cloud water. The local value of $r_c$ decreases, which decreases $r_t$ and increases $\theta_l$ preferentially in the moistest portions of domain. As a result, scalar variances $\overline{r_t'^2}$ and $\overline{\theta_l'^2}$ are reduced, and the (negative) covariance $\overline{r_t'\theta_l'}$ is reduced in magnitude. Similarly, since moister regions of cloud are associated with stronger updrafts, the covariance $\overline{w'r_t'}$ is reduced by microphysics and the (negative) covariance $\overline{w'\theta_l'}$ is reduced in magnitude by microphysics.

In the region below cloud, a different microphysical process occurs: rain falls into clear air below cloud and evaporates. Evaporation increases water vapor at the expense of rain water and also cools the air. Hence, where evaporation occurs, $r_t$ is increased and $\theta_l$ is decreased. If rain preferentially falls through regions of air that have already been cooled by evaporation, then cool air is further cooled. In a partly rainy case such as RICO, rain cools the rainshafts but not other portions of the domain, increasing variability in $\theta_l$. In RICO, the positive tendency of subcloud $\overline{\theta_l'^2}$ by microphysics is significant, as shown in Fig. 4. Parameterizing a positive subcloud microphysics tendency of $\overline{\theta_l'^2}$ in CLUBB requires prescribing the within-component correlations such that rain tends to fall in cool air below cloud. The fact that CLUBB is able to parameterize this effect (Fig. 4(b)) opens the door to future parameterization of the effects of cold pools on convection.

## 4.2 DYCOMS-II RF02 drizzling stratocumulus

The aforementioned results show that CLUBB's formulas are able to qualitatively approximate the microphysical (co)variance terms for a boundary-layer case with small cloud fraction (RICO). Can CLUBB's formulas approximate the microphysical (co)variance terms produced by SAM LES in other cloud cases? To begin to address this question, we simulate a boundary-layer case that has a layer-averaged rain water mixing ratio that is comparable to that of RICO but that has a cloud fraction and precipitation fraction of nearly one. The case we simulate is the DYCOMS-II RF02 marine stratocumulus case.

In the SAM LES budgets of DYCOMS-II RF02 for $\overline{r_t'^2}$ (Fig. 6(a)) and $\overline{\theta_l'^2}$ (Fig. 7(a)), the microphysics term is negligible in comparison to the other terms. While CLUBB's dissipation and production terms are overestimated, CLUBB's microphysics term is also negligible for both $\overline{r_t'^2}$ (Fig. 6(b)) and $\overline{\theta_l'^2}$ (Fig. 7(b)), in agreement with SAM LES. In addition, both SAM and CLUBB show similarly negligible microphysics terms in the $\overline{w'r_t'}$, $\overline{w'\theta_l'}$, and $\overline{r_t'\theta_l'}$ budgets (not shown). This agreement suggests that CLUBB's microphysical (co)variance formulas are applicable for shallow cloud cases with either small or large values of cloud fraction.

Why are the microphysics budget terms so much less significant in DYCOMS-II RF02 than they are in RICO? Consider the covariance of a field and a microphysics process rate — for example, the covariance of $\theta_l$ and accretion rate. The magnitude of this covariance is related, in part, to the magnitude of $\overline{\theta_l'^2}$ and the magnitude of the variance of accretion rate. (We set aside the issue of the correlation between the two fields.) Comparing SAM LES results at altitudes where precipitation is large, both $\overline{\theta_l'^2}$ and $\overline{r_t'^2}$ are smaller in DYCOMS-II RF02 than in RICO (not shown). This is because marine stratocumulus cloud layers are well mixed by turbulence. DYCOMS-II RF02 also exhibits less variability in microphysical process rates. The variance of warm-rain microphysics process rates is related, in part, to the variance of rain water mixing ratio. In RICO, precipitation is found over a small region of the horizontal domain, while in the overcast DYCOMS-II RF02 case, precipitation is found over almost the entire horizontal domain. The in-precipitation mean of $r_r$ is much larger in RICO than it is in DYCOMS-II RF02 (not shown). Additionally, in RICO the ratio of the *in-precipitation* variance to the square of the *in-precipitation* mean for $r_r$ ($\approx 5$) is much larger than the corresponding ratio ($\lesssim 1$) in DYCOMS-II RF02. As a result, both the in-precipitation variance and the layer-mean variance of $r_r$ is much greater in RICO than in DYCOMS-II RF02. In summary, the microphysical (co)variance terms are smaller in marine stratocumuli than in cumuli partly because both the thermodynamic and microphysical fields are more homogeneous in marine stratocumuli.

## 4.3 RICO sensitivity study: How significant are the microphysical (co)variance terms?

If the microphysical terms in the (co)variance equations are omitted, how large are the resulting errors? To address this, a second CLUBB simulation of RICO was run that is identical to the original simulation with the one exception that the microphysical (co)variance terms are turned off.

When the microphysical (co)variance terms are removed, compensating errors in other terms must be induced in order to restore balance in the budgets. A large compensation occurs in the budgets for the scalar (co)variances because the microphysical terms in those budgets are large. The budgets of scalar variances $\overline{r_t'^2}$ and $\overline{\theta_l'^2}$ are shown for the CLUBB simulation with

microphysics feedback disabled in Fig. 8(a) and Fig. 8(b), respectively. When compared to the same budgets from the CLUBB simulation with microphysics feedback enabled in Fig. 3(b) and Fig. 4(b), respectively, the terms from the simulation with microphysics feedback disabled are all much larger in magnitude. Within the cloudy layer, microphysics is a dominant sink of scalar variances. In order to compensate for the loss of that sink term, both dissipation and advection (below 3000 m) increase in (negative) magnitude. Since the integral of the (turbulent) advection term over the vertical profile must have a mass-weighted vertical integral of $\approx 0$, it becomes an excessive source of $\overline{r_t'^2}$ and $\overline{\theta_l'^2}$ above 3000 m. In essence, when the microphysical sink of variance is removed, the layer becomes more variable, develops more turbulence, and grows deeper. Similar characteristics are exhibited in the budget of $\overline{r_t'\theta_l'}$ (not shown).

The budgets of turbulent fluxes $\overline{w'r_t'}$ and $\overline{w'\theta_l'}$ are shown for the CLUBB simulation with microphysics terms disabled in Fig. 8(c) and Fig. 8(d), respectively. When compared to the same budgets from the CLUBB simulation with microphysics terms enabled in Fig. 1(b) and Fig. 2(b), respectively, the buoyancy+pressure terms, the advection terms, and the $\overline{w'r_t'}$ production term all increase in magnitude. This increase is relatively small, however, which is expected because microphysics is a less significant term in the turbulent flux budgets. The terms from the simulation with microphysics terms disabled extend much higher in altitude, again because the layer has more vigorous turbulence.

The errors induced by the loss of the microphysical (co)variance terms propagate throughout the model solution, infecting, for instance, the mean fields. Figure 9 shows profiles of mean fields in a three-way comparison between 1) SAM LES, 2) CLUBB with microphysical effects on (co)variances disabled, and 3) CLUBB with microphysical effects on (co)variances enabled. In Fig. 9(a), when the microphysical effects on (co)variances are turned off, CLUBB's $\overline{\theta_l}$ becomes too warm at lower altitudes and too cool aloft. As a result, Fig. 9(c) shows that $\overline{r_c}$ extends too high in altitude when compared to SAM LES. Omitting the microphysical (co)variance terms would constitute a significant model error.

## 5 Conclusions

Microphysical sources of (co)variances of total water and liquid water potential temperature are significant. A LES of the RICO shallow cumulus case shows that, in this cloud case, microphysical sources are major terms in the budgets of variances and turbulent fluxes. In particular, microphysical processes have three main effects. First, precipitation formation and growth is the major sink of $\overline{r_t'^2}$, $\overline{\theta_l'^2}$, and the magnitude of $\overline{r_t'\theta_l'}$ in the upper half of the cloud layer (see Figs. 3, 4, and 5). In particular, microphysical damping is greater than turbulent dissipation. The damping of scalar variances occurs because rain formation depletes cloud water preferentially in the moistest part of the cloud. This depletion preferentially reduces the largest values of (liquid) cloud water, thereby reducing the horizontally-averaged variance. Second, microphysics also damps the turbulent flux of scalars, $\overline{w'r_t'}$ and $\overline{w'\theta_l'}$ (see Figs. 1 and 2). The mechanism is the same: precipitation reduces cloud water in the moistest part of the cloud, which also contains stronger updrafts. Although the effects of microphysics on fluxes are smaller than those on variances, microphysics is still a major term in the $\overline{w'\theta_l'}$ budget and ought not to be ignored. Third, evaporation of rain below cloud acts as a source of $\overline{\theta_l'^2}$. The positive sign arises because evaporation of rain cools the cooler part of the subcloud layer.

This evaporation-induced generation of $\overline{\theta_l'^2}$ is a key aspect of cold pool formation. It leads to buoyant generation of $\overline{w'\theta_l'}$ below cloud base, which in turn leads to new convection.

This paper demonstrates that all these microphysical sources and sinks can be calculated analytically, given a sufficiently simple warm-rain microphysics scheme and a sufficiently simple multivariate PDF. These analytic expressions have been im-
plemented in the predictive equations for variances and covariances involving $r_t$ and $\theta_l$ in the CLUBB parameterization. When applied in an interactive, single-column simulation of the RICO case by CLUBB, the microphysical terms agree qualitatively with LES in sign and in relative magnitude .

In the future, analytic integration of microphysical sources of scalar (co)variances may provide a useful step for the parameterization of cold pools and cloud organization. It does not parameterize cold pools and cloud organization directly, because it does not account for spatial arrangement of cloud parcels. Furthermore, it does not even parameterize all *effects* of cold pools and cloud organization. However, it does parameterize effects that are directly related to scalar variability, and it parameterizes these effects in a non-phenomenological, rigorous way. Namely, it defines the microphysical sources with precise, mathematical expressions, and it provides explicit formulas for the case of idealized, warm-rain microphysics. Although the effects of cold pools are relatively modest in the statistically steady, shallow-cumulus case analyzed in this paper, the effects are larger in some transient, deep convective cases (e.g., Khairoutdinov and Randall, 2002, 2006).

In addition, analytic integration assists in the development of more general integration methods, such as Monte Carlo integration (Larson and Schanen, 2013). For instance, analytic integration allows a researcher to rapidly explore behaviors in idealized settings while avoiding the contamination of sampling noise or other integration errors. More importantly, analytic integration provides an alternative solution that can be used to test whether a Monte Carlo integration code converges to the correct solution (Larson and Schanen, 2013). In past experience, we have found such testing to be crucial. Bugs are surprisingly easy to introduce, and without comparison against an independent solution, results produced by a Monte Carlo integrator will be subject to lingering doubts. On the other hand, once a Monte Carlo integrator has been tested against an analytic solution, it can be used more confidently with a comprehensive microphysics scheme that includes ice in order to simulate a variety of shallow and deep cloud cases. In fact, this has already been done in Storer et al. (2015). In this way, analytic integration of the microphysical effects on scalar variances and fluxes is an enabling technology: it enables the verification of general subgrid integration methods.

## Appendix A: Covariances involving microphysics process rates

This Appendix sets up the integrals that need to be solved in order to find the microphysical covariance terms listed in Section 2.1. The integrals set up here can be evaluated using the expressions given in the Supplement.

The nine microphysical covariances involving each of $w$, $r_t$, and $\theta_l$ with each of KK autoconversion rate, accretion rate, and evaporation rate are calculated by integrating over the PDF. The KK microphysics process rates are calculated, in part, based on variables that involve saturation, such as $r_c$. In order to calculate quantities that involve saturation, a PDF transformation, which is a change of coordinates, is required. The multivariate PDF undergoes stretching, translation, and rotation of the axes

(Larson et al., 2005; Mellor, 1977). An independent PDF transformation takes place in each PDF component. Ultimately, $r_t$ and $\theta_l$ are replaced in the PDF by $\chi$ and $\eta$, where $\chi$ is an "extended" liquid water mixing ratio that has a positive value when air is supersaturated. In this scenario, $\chi$ is also equal to $r_c$. When air is subsaturated, $\chi$ has a negative value. The variable $\eta$ is orthogonal to $\chi$. The transformations that relate $r_t$ and $\theta_l$ to $\chi$ and $\eta$ are

$$c_{r_t(i)}\left(r_t - \mu_{r_t(i)}\right) = \frac{\left(\eta - \mu_{\eta(i)}\right) + \left(\chi - \mu_{\chi(i)}\right)}{2}, \text{ and} \tag{A1}$$

$$c_{\theta_l(i)}\left(\theta_l - \mu_{\theta_l(i)}\right) = \frac{\left(\eta - \mu_{\eta(i)}\right) - \left(\chi - \mu_{\chi(i)}\right)}{2}, \tag{A2}$$

where $\mu_{r_t(i)}$ is the mean of $r_t$ in the $i$th PDF component and $\mu_{\theta_l(i)}$ is the mean of $\theta_l$ in the $i$th PDF component.

The mean of $\chi$ in the $i$th PDF component, $\mu_{\chi(i)}$, is given by

$$\mu_{\chi(i)} = \frac{\mu_{r_t(i)} - r_{sw}\left(\mu_{T_l(i)}, p\right)}{1 + \Lambda\left(\mu_{T_l(i)}\right) r_{sw}\left(\mu_{T_l(i)}, p\right)}, \tag{A3}$$

where $r_{sw}\left(\mu_{T_l(i)}, p\right)$ is the saturation mixing ratio with respect to liquid water, $\mu_{T_l(i)}$ is the mean of $T_l$ in the $i$th PDF component, and $\Lambda\left(\mu_{T_l(i)}\right)$ is given by

$$\Lambda\left(\mu_{T_l(i)}\right) = \frac{R_d}{R_v}\left(\frac{L_v}{R_d \mu_{T_l(i)}}\right)\left(\frac{L_v}{C_{pd} \mu_{T_l(i)}}\right), \tag{A4}$$

where $R_v$ is the gas constant for water vapor. The mean of $\eta$ in the $i$th PDF component, $\mu_{\eta(i)}$, ultimately does not factor into the solution to the integral equations. Its value is irrelevant and can be set to an arbitrary value, such as 0, for simplicity. However, it should be noted that the PDF component standard deviations of $\eta$ and PDF component correlations involving $\eta$ still factor into the solution. The coefficients $c_{r_t(i)}$ and $c_{\theta_l(i)}$ are given by

$$c_{r_t(i)} = \frac{1}{1 + \Lambda\left(\mu_{T_l(i)}\right) r_{sw}\left(\mu_{T_l(i)}, p\right)}, \text{ and} \tag{A5}$$

$$c_{\theta_l(i)} = \frac{\left(1 + \Lambda\left(\mu_{T_l(i)}\right) \mu_{r_t(i)}\right) \Lambda\left(\mu_{T_l(i)}\right) r_{sw}\left(\mu_{T_l(i)}, p\right)}{\left(1 + \Lambda\left(\mu_{T_l(i)}\right) r_{sw}\left(\mu_{T_l(i)}, p\right)\right)^2} \frac{C_{pd}}{L_v}\left(\frac{\overline{p}}{p_0}\right)^{\frac{R_d}{C_{pd}}}. \tag{A6}$$

**A1   Covariances involving autoconversion rate**

The general form of the KK equation for autoconversion rate is the product of a coefficient and $r_c^\alpha N_c^\beta$ (where for KK, $\alpha = 2.47$ and $\beta = -1.79$). The integral equation for the covariance of $w$ and autoconversion rate involves the PDF-variables $w$, $r_t$, $\theta_l$, and $N_{cn}$. The equation is

$$\overline{w' \frac{\partial r_r}{\partial t}\bigg|'_{\text{auto}}} = \int\limits_{-\infty}^{\infty} \int\limits_{-\infty}^{\infty} \int\limits_{-\infty}^{\infty} \int\limits_{0}^{\infty} \left(w - \overline{w}\right) \left(\frac{\partial r_r}{\partial t}\bigg|_{\text{auto}} - \overline{\frac{\partial r_r}{\partial t}\bigg|_{\text{auto}}}\right)$$

$$\times P\left(w, r_t, \theta_l, N_{cn}\right) \mathrm{d}N_{cn}\, \mathrm{d}\theta_l\, \mathrm{d}r_t\, \mathrm{d}w. \tag{A7}$$

The PDF is transformed (in each component) from $r_t$ and $\theta_l$ coordinates to $\chi$ and $\eta$ coordinates. Additionally, $r_c = \chi H(\chi)$ and $N_c = N_{cn} H(\chi)$, where $H(\chi)$ is the Heaviside step function (Griffin and Larson, 2016). The equation becomes

$$
\overline{w'\left.\frac{\partial r_r}{\partial t}\right|'_{\text{auto}}} = \sum_{i=1}^{n} \xi_{(i)} \int_{-\infty}^{\infty} \int_{-\infty}^{\infty} \int_{-\infty}^{\infty} \int_{0}^{\infty} \left( w - \overline{w} \right) \left( C_{\text{auto}} \chi^{\alpha} N_{cn}^{\beta} \left( H(\chi) \right)^{\alpha+\beta} - \overline{\left.\frac{\partial r_r}{\partial t}\right|_{\text{auto}}} \right)
$$

$$
\times\, P_{(i)}\left(w, \chi, \eta, N_{cn}\right) \mathrm{d}N_{cn}\, \mathrm{d}\eta\, \mathrm{d}\chi\, \mathrm{d}w, \tag{A8}
$$

where the coefficient $C_{\text{auto}} = 1350 \left(10^{-6} \rho_d\right)^{\beta}$, and where $\rho_d$ is the density of dry air. The variable $\eta$ can be integrated out of the PDF. The equation for the covariance of $w$ and autoconversion rate is

$$
\overline{w'\left.\frac{\partial r_r}{\partial t}\right|'_{\text{auto}}}
$$

$$
= C_{\text{auto}} \sum_{i=1}^{n} \xi_{(i)} \int_{-\infty}^{\infty} \int_{-\infty}^{\infty} \int_{0}^{\infty} \left( w - \overline{w} \right) \left( \chi^{\alpha} N_{cn}^{\beta} \left( H(\chi) \right)^{\alpha+\beta} - \frac{1}{C_{\text{auto}}} \overline{\left.\frac{\partial r_r}{\partial t}\right|_{\text{auto}}} \right)
$$

$$
\times\, P_{NNL(i)}\left(w, \chi, N_{cn}\right) \mathrm{d}N_{cn}\, \mathrm{d}\chi\, \mathrm{d}w, \tag{A9}
$$

where $P_{NNL(i)}\left(w, \chi, N_{cn}\right)$ is the $i$th component trivariate PDF involving two normal variates and one lognormal variate. The functional form of the PDF (for the $i$th PDF component) is given in the Supplement in Eq. (S2), and the integral is solved (for the $i$ PDF component) in Section S6 (Eq. (S25) through Eq. (S32)).

The integral equation for the covariance of $r_t$ and autoconversion rate involves the PDF-variables $r_t$, $\theta_l$, and $N_{cn}$. The equation is

$$
\overline{r_t'\left.\frac{\partial r_r}{\partial t}\right|'_{\text{auto}}} = \int_{-\infty}^{\infty} \int_{-\infty}^{\infty} \int_{0}^{\infty} \left( r_t - \overline{r_t} \right) \left( \left.\frac{\partial r_r}{\partial t}\right|_{\text{auto}} - \overline{\left.\frac{\partial r_r}{\partial t}\right|_{\text{auto}}} \right) P\left(r_t, \theta_l, N_{cn}\right) \mathrm{d}N_{cn}\, \mathrm{d}\theta_l\, \mathrm{d}r_t. \tag{A10}
$$

During the PDF transformation, Eq. (A1) is used to substitute for $r_t$. The equation becomes

$$
\overline{r_t'\left.\frac{\partial r_r}{\partial t}\right|'_{\text{auto}}} = \sum_{i=1}^{n} \xi_{(i)} \int_{-\infty}^{\infty} \int_{-\infty}^{\infty} \int_{0}^{\infty} \left( \mu_{r_t(i)} - \overline{r_t} + \frac{\left(\eta - \mu_{\eta(i)}\right) + \left(\chi - \mu_{\chi(i)}\right)}{2 c_{r_t(i)}} \right)
$$

$$
\times \left( C_{\text{auto}} \chi^{\alpha} N_{cn}^{\beta} \left( H(\chi) \right)^{\alpha+\beta} - \overline{\left.\frac{\partial r_r}{\partial t}\right|_{\text{auto}}} \right)
$$

$$
\times\, P_{(i)}\left(\eta, \chi, N_{cn}\right) \mathrm{d}N_{cn}\, \mathrm{d}\chi\, \mathrm{d}\eta. \tag{A11}
$$

The integral equation is split and simplified, and becomes

$$
\overline{r'_t \left.\frac{\partial r_r}{\partial t}\right|'_{\text{auto}}}
$$

$$
= C_{\text{auto}} \sum_{i=1}^{n} \xi_{(i)}
$$

$$
\times \left( \frac{1}{2c_{r_t(i)}} \int\limits_{-\infty}^{\infty} \int\limits_{-\infty}^{\infty} \int\limits_{0}^{\infty} \left( \eta - \mu_{\eta(i)} \right) \left( \chi^{\alpha} N_{cn}^{\beta} \left( H\left(\chi\right)\right)^{\alpha+\beta} - \frac{1}{C_{\text{auto}}} \overline{\left.\frac{\partial r_r}{\partial t}\right|_{\text{auto}}} \right) \right.
$$

$$
\times P_{NNL(i)} \left( \eta, \chi, N_{cn} \right) \mathrm{d}N_{cn} \, \mathrm{d}\chi \, \mathrm{d}\eta
$$

$$
+ \frac{1}{2c_{r_t(i)}} \int\limits_{0}^{\infty} \int\limits_{0}^{\infty} \chi^{\alpha+1} N_{cn}^{\beta} P_{NL(i)} \left( \chi, N_{cn} \right) \mathrm{d}N_{cn} \, \mathrm{d}\chi
$$

$$
+ \left( \mu_{r_t(i)} - \overline{r_t} - \frac{\mu_{\chi(i)}}{2c_{r_t(i)}} \right) \int\limits_{0}^{\infty} \int\limits_{0}^{\infty} \chi^{\alpha} N_{cn}^{\beta} P_{NL(i)} \left( \chi, N_{cn} \right) \mathrm{d}N_{cn} \, \mathrm{d}\chi \Bigg), \tag{A12}
$$

where $P_{NL(i)}\left(\chi, N_{cn}\right)$ is the $i$th component bivariate PDF involving one normal variate and one lognormal variate. The functional form of the trivariate NNL PDF (for the $i$th PDF component) is given in the Supplement in Eq. (S2), and the related integral is solved (for the $i$ PDF component) in Section S6 (Eq. (S25) through Eq. (S32)). The functional form of the bivariate NL PDF (for the $i$th PDF component) is given in Eq. (S5), and the related integrals are solved (for the $i$th PDF component) by using the general form given in Section S8 (Eq. (S41) through Eq. (S44)).

The integral equation for the covariance of $\theta_l$ and autoconversion rate involves the PDF-variables $r_t$, $\theta_l$, and $N_{cn}$. The equation is

$$
\overline{\theta'_l \left.\frac{\partial r_r}{\partial t}\right|'_{\text{auto}}} = \int\limits_{-\infty}^{\infty} \int\limits_{-\infty}^{\infty} \int\limits_{0}^{\infty} \left( \theta_l - \overline{\theta_l} \right) \left( \left.\frac{\partial r_r}{\partial t}\right|_{\text{auto}} - \overline{\left.\frac{\partial r_r}{\partial t}\right|_{\text{auto}}} \right) P\left( r_t, \theta_l, N_{cn} \right) \mathrm{d}N_{cn} \, \mathrm{d}\theta_l \, \mathrm{d}r_t. \tag{A13}
$$

During the PDF transformation, Eq. (A2) is used to substitute for $\theta_l$. The equation becomes

$$
\overline{\theta'_l \left.\frac{\partial r_r}{\partial t}\right|'_{\text{auto}}} = \sum_{i=1}^{n} \xi_{(i)} \int\limits_{-\infty}^{\infty} \int\limits_{-\infty}^{\infty} \int\limits_{0}^{\infty} \left( \mu_{\theta_l(i)} - \overline{\theta_l} + \frac{\left( \eta - \mu_{\eta(i)} \right) - \left( \chi - \mu_{\chi(i)} \right)}{2c_{\theta_l(i)}} \right)
$$

$$
\times \left( C_{\text{auto}} \chi^{\alpha} N_{cn}^{\beta} \left( H\left(\chi\right)\right)^{\alpha+\beta} - \overline{\left.\frac{\partial r_r}{\partial t}\right|_{\text{auto}}} \right)
$$

$$
\times P_{(i)} \left( \eta, \chi, N_{cn} \right) \mathrm{d}N_{cn} \, \mathrm{d}\chi \, \mathrm{d}\eta. \tag{A14}
$$

The integral equation is split and simplified, and becomes

$$
\overline{\theta_l' \left.\frac{\partial r_r}{\partial t}\right|'_{\text{auto}}}
$$

$$
= C_{\text{auto}} \sum_{i=1}^{n} \xi_{(i)}
$$

$$
\times \left( \frac{1}{2c_{\theta_l(i)}} \int\limits_{-\infty}^{\infty} \int\limits_{-\infty}^{\infty} \int\limits_{0}^{\infty} \left( \eta - \mu_{\eta(i)} \right) \left( \chi^\alpha N_{cn}^\beta \left( H\left(\chi\right) \right)^{\alpha+\beta} - \frac{1}{C_{\text{auto}}} \overline{\left.\frac{\partial r_r}{\partial t}\right|_{\text{auto}}} \right) \right.
$$

$$
\times P_{NNL(i)} \left( \eta, \chi, N_{cn} \right) \mathrm{d}N_{cn}\, \mathrm{d}\chi\, \mathrm{d}\eta
$$

$$
- \frac{1}{2c_{\theta_l(i)}} \int\limits_{0}^{\infty} \int\limits_{0}^{\infty} \chi^{\alpha+1} N_{cn}^\beta P_{NL(i)} \left( \chi, N_{cn} \right) \mathrm{d}N_{cn}\, \mathrm{d}\chi
$$

$$
\left. + \left( \mu_{\theta_l(i)} - \overline{\theta_l} + \frac{\mu_{\chi(i)}}{2c_{\theta_l(i)}} \right) \int\limits_{0}^{\infty} \int\limits_{0}^{\infty} \chi^\alpha N_{cn}^\beta P_{NL(i)} \left( \chi, N_{cn} \right) \mathrm{d}N_{cn}\, \mathrm{d}\chi \right). \tag{A15}
$$

The functional form of the trivariate NNL PDF (for the $i$th PDF component) is given in the Supplement in Eq. (S2), and the related integral is solved (for the $i$ PDF component) in Section S6 (Eq. (S25) through Eq. (S32)). The functional form of the bivariate NL PDF (for the $i$th PDF component) is given in Eq. (S5), and the related integrals are solved (for the $i$th PDF component) by using the general form given in Section S8 (Eq. (S41) through Eq. (S44)).

## A2 Covariances involving accretion rate

The general form of the KK equation for accretion rate is the product of a coefficient and $r_c^\alpha r_r^\beta$ (where for KK, $\alpha = 1.15$ and $\beta = 1.15$). The integral equation for the covariance of $w$ and accretion rate involves the PDF-variables $w$, $r_t$, $\theta_l$, and $r_r$. The equation is

$$
\overline{w' \left.\frac{\partial r_r}{\partial t}\right|'_{\text{accr}}} = \int\limits_{-\infty}^{\infty} \int\limits_{-\infty}^{\infty} \int\limits_{-\infty}^{\infty} \int\limits_{0}^{\infty} \left( w - \overline{w} \right) \left( \left.\frac{\partial r_r}{\partial t}\right|_{\text{accr}} - \overline{\left.\frac{\partial r_r}{\partial t}\right|_{\text{accr}}} \right)
$$

$$
\times P\left( w, r_t, \theta_l, r_r \right) \mathrm{d}r_r\, \mathrm{d}\theta_l\, \mathrm{d}r_t\, \mathrm{d}w. \tag{A16}
$$

The PDF is transformed and the substitution $r_c = \chi H\left(\chi\right)$ is made. The equation becomes

$$
\overline{w' \left.\frac{\partial r_r}{\partial t}\right|'_{\text{accr}}} = \sum_{i=1}^{n} \xi_{(i)} \int\limits_{-\infty}^{\infty} \int\limits_{-\infty}^{\infty} \int\limits_{-\infty}^{\infty} \int\limits_{0}^{\infty} \left( w - \overline{w} \right) \left( C_{\text{accr}} \chi^\alpha \left( H\left(\chi\right) \right)^\alpha r_r^\beta - \overline{\left.\frac{\partial r_r}{\partial t}\right|_{\text{accr}}} \right)
$$

$$
\times P_{(i)} \left( w, \chi, \eta, r_r \right) \mathrm{d}r_r\, \mathrm{d}\eta\, \mathrm{d}\chi\, \mathrm{d}w, \tag{A17}
$$

where the coefficient $C_{\text{accr}} = 67$. The variable $\eta$ is integrated out of the PDF, and the equation for the covariance of $w$ and accretion rate is

$$
\overline{w'\left.\frac{\partial r_r}{\partial t}\right|'_{\text{accr}}} = C_{\text{accr}} \sum_{i=1}^{n} \xi_{(i)}
$$
$$
\times \left( f_{p(i)} \int_{-\infty}^{\infty} \int_{-\infty}^{\infty} \int_{0}^{\infty} (w - \overline{w}) \left( \chi^{\alpha} \left( H\left(\chi\right) \right)^{\alpha} r_r^{\beta} - \frac{1}{C_{\text{accr}}} \overline{\left.\frac{\partial r_r}{\partial t}\right|_{\text{accr}}} \right) \right.
$$
$$
\times P_{NNL(i)}\left(w, \chi, r_r\right) \mathrm{d}r_r \,\mathrm{d}\chi \,\mathrm{d}w
$$
$$
\left. - \left(1 - f_{p(i)}\right) \left(\mu_{w(i)} - \overline{w}\right) \frac{1}{C_{\text{accr}}} \overline{\left.\frac{\partial r_r}{\partial t}\right|_{\text{accr}}} \right),
\tag{A18}
$$

where $\mu_{w(i)}$ is the mean of $w$ in the $i$th PDF component. The functional form of the PDF (for the $i$th PDF component) is given in the Supplement in Eq. (S2), and the integral is solved (for the $i$ PDF component) in Section S6 (Eq. (S25) through Eq. (S32)).

The integral equation for the covariance of $r_t$ and accretion rate involves the PDF-variables $r_t$, $\theta_l$, and $r_r$. The equation is

$$
\overline{r'_t \left.\frac{\partial r_r}{\partial t}\right|'_{\text{accr}}} = \int_{-\infty}^{\infty} \int_{-\infty}^{\infty} \int_{0}^{\infty} \left(r_t - \overline{r_t}\right) \left( \overline{\left.\frac{\partial r_r}{\partial t}\right|_{\text{accr}}} - \overline{\left.\frac{\partial r_r}{\partial t}\right|_{\text{accr}}} \right) P\left(r_t, \theta_l, r_r\right) \mathrm{d}r_r \,\mathrm{d}\theta_l \,\mathrm{d}r_t.
\tag{A19}
$$

The PDF is transformed (in each component) from $r_t$ and $\theta_l$ coordinates to $\chi$ and $\eta$ coordinates. The equation becomes

$$
\overline{r'_t \left.\frac{\partial r_r}{\partial t}\right|'_{\text{accr}}} = \sum_{i=1}^{n} \xi_{(i)} \int_{-\infty}^{\infty} \int_{-\infty}^{\infty} \int_{0}^{\infty} \left( \mu_{r_t(i)} - \overline{r_t} + \frac{\left(\eta - \mu_{\eta(i)}\right) + \left(\chi - \mu_{\chi(i)}\right)}{2 c_{r_t(i)}} \right)
$$
$$
\times \left( C_{\text{accr}} \chi^{\alpha} \left(H\left(\chi\right)\right)^{\alpha} r_r^{\beta} - \overline{\left.\frac{\partial r_r}{\partial t}\right|_{\text{accr}}} \right)
$$
$$
\times P_{(i)}\left(\eta, \chi, r_r\right) \mathrm{d}r_r \,\mathrm{d}\chi \,\mathrm{d}\eta.
\tag{A20}
$$

The equation for the covariance of $r_t$ and accretion rate can ultimately be written as

$$
\overline{r'_t \left.\frac{\partial r_r}{\partial t}\right|'_{\text{accr}}} = C_{\text{accr}} \sum_{i=1}^{n} \xi_{(i)} f_{p(i)}
$$
$$
\times \left( \frac{1}{2 c_{r_t(i)}} \int_{-\infty}^{\infty} \int_{-\infty}^{\infty} \int_{0}^{\infty} \left(\eta - \mu_{\eta(i)}\right) \left( \chi^{\alpha} \left(H\left(\chi\right)\right)^{\alpha} r_r^{\beta} - \frac{1}{C_{\text{accr}}} \overline{\left.\frac{\partial r_r}{\partial t}\right|_{\text{accr}}} \right) \right.
$$
$$
\times P_{NNL(i)}\left(\eta, \chi, r_r\right) \mathrm{d}r_r \,\mathrm{d}\chi \,\mathrm{d}\eta
$$
$$
+ \frac{1}{2 c_{r_t(i)}} \int_{0}^{\infty} \int_{0}^{\infty} \chi^{\alpha+1} r_r^{\beta} P_{NL(i)}\left(\chi, r_r\right) \mathrm{d}r_r \,\mathrm{d}\chi
$$
$$
\left. + \left(\mu_{r_t(i)} - \overline{r_t} - \frac{\mu_{\chi(i)}}{2 c_{r_t(i)}}\right) \int_{0}^{\infty} \int_{0}^{\infty} \chi^{\alpha} r_r^{\beta} P_{NL(i)}\left(\chi, r_r\right) \mathrm{d}r_r \,\mathrm{d}\chi \right).
\tag{A21}
$$

The functional form of the trivariate NNL PDF (for the $i$th PDF component) is given in the Supplement in Eq. (S2), and the related integral is solved (for the $i$ PDF component) in Section S6 (Eq. (S25) through Eq. (S32)). The functional form of the bivariate NL PDF (for the $i$th PDF component) is given in Eq. (S5), and the related integrals are solved (for the $i$th PDF component) by using the general form given in Section S8 (Eq. (S41) through Eq. (S44)).

5    The integral equation for the covariance of $\theta_l$ and accretion rate involves the PDF-variables $r_t$, $\theta_l$, and $r_r$. The equation is

$$\overline{\theta_l' \left.\frac{\partial r_r}{\partial t}\right|_{\mathrm{accr}}}{}' = \int_{-\infty}^{\infty} \int_{-\infty}^{\infty} \int_{0}^{\infty} \left( \theta_l - \overline{\theta_l} \right) \left( \left.\frac{\partial r_r}{\partial t}\right|_{\mathrm{accr}} - \overline{\left.\frac{\partial r_r}{\partial t}\right|_{\mathrm{accr}}} \right) P\left(r_t, \theta_l, r_r\right) \mathrm{d}r_r \, \mathrm{d}\theta_l \, \mathrm{d}r_t. \tag{A22}$$

A PDF transformation takes place in each component, changing coordinates from $r_t$ and $\theta_l$ to $\chi$ and $\eta$. The equation becomes

$$\overline{\theta_l' \left.\frac{\partial r_r}{\partial t}\right|_{\mathrm{accr}}}{}' = \sum_{i=1}^{n} \xi_{(i)} \int_{-\infty}^{\infty} \int_{-\infty}^{\infty} \int_{0}^{\infty} \left( \mu_{\theta_l(i)} - \overline{\theta_l} + \frac{\left(\eta - \mu_{\eta(i)}\right) - \left(\chi - \mu_{\chi(i)}\right)}{2c_{\theta_l(i)}} \right)$$
$$\times \left( C_{\mathrm{accr}} \chi^\alpha \left(H\left(\chi\right)\right)^\alpha r_r^\beta - \overline{\left.\frac{\partial r_r}{\partial t}\right|_{\mathrm{accr}}} \right)$$
$$\times P_{(i)}\left(\eta, \chi, r_r\right) \mathrm{d}r_r \, \mathrm{d}\chi \, \mathrm{d}\eta. \tag{A23}$$

The equation for the covariance of $\theta_l$ and accretion rate can ultimately be written as

$$\overline{\theta_l' \left.\frac{\partial r_r}{\partial t}\right|_{\mathrm{accr}}}{}' = C_{\mathrm{accr}} \sum_{i=1}^{n} \xi_{(i)} f_{p(i)}$$
$$\times \left( \frac{1}{2c_{\theta_l(i)}} \int_{-\infty}^{\infty} \int_{-\infty}^{\infty} \int_{0}^{\infty} \left( \eta - \mu_{\eta(i)} \right) \left( \chi^\alpha \left(H\left(\chi\right)\right)^\alpha r_r^\beta - \frac{1}{C_{\mathrm{accr}}} \overline{\left.\frac{\partial r_r}{\partial t}\right|_{\mathrm{accr}}} \right) \right.$$
$$\times P_{NNL(i)}\left(\eta, \chi, r_r\right) \mathrm{d}r_r \, \mathrm{d}\chi \, \mathrm{d}\eta$$
$$- \frac{1}{2c_{\theta_l(i)}} \int_{0}^{\infty} \int_{0}^{\infty} \chi^{\alpha+1} r_r^\beta P_{NL(i)}\left(\chi, r_r\right) \mathrm{d}r_r \, \mathrm{d}\chi$$
$$\left. + \left( \mu_{\theta_l(i)} - \overline{\theta_l} + \frac{\mu_{\chi(i)}}{2c_{\theta_l(i)}} \right) \int_{0}^{\infty} \int_{0}^{\infty} \chi^\alpha r_r^\beta P_{NL(i)}\left(\chi, r_r\right) \mathrm{d}r_r \, \mathrm{d}\chi \right). \tag{A24}$$

The functional form of the trivariate NNL PDF (for the $i$th PDF component) is given in the Supplement in Eq. (S2), and the related integral is solved (for the $i$ PDF component) in Section S6 (Eq. (S25) through Eq. (S32)). The functional form of the bivariate NL PDF (for the $i$th PDF component) is given in Eq. (S5), and the related integrals are solved (for the $i$th PDF component) by using the general form given in Section S8 (Eq. (S41) through Eq. (S44)).

## 15  A3   Covariances involving evaporation rate

The general form of the KK equation for evaporation rate is the product of a coefficient and $S^\alpha \left(H\left(-S\right)\right)^\alpha r_r^\beta N_r^\gamma$ (where for KK, $\alpha = 1$, $\beta = 1/3$, and $\gamma = 2/3$). Supersaturation, $S$, is the ratio of water vapor pressure over saturation vapor pressure (with

respect to liquid water), minus 1, so that $S$ is positive when air is supersaturated and negative when air is subsaturated. The Heaviside step function has been added to allow for only evaporation of rain, not condensational growth. The integral equation for the covariance of $w$ and evaporation rate involves the PDF-variables $w$, $r_t$, $\theta_l$, $r_r$, and $N_r$. The equation is

$$\overline{w'\left.\frac{\partial r_r}{\partial t}\right|'_{\text{evap}}} = \int\limits_{-\infty}^{\infty}\int\limits_{-\infty}^{\infty}\int\limits_{-\infty}^{\infty}\int\limits_{0}^{\infty}\int\limits_{0}^{\infty} (w-\overline{w})\left(\left.\frac{\partial r_r}{\partial t}\right|_{\text{evap}} - \overline{\left.\frac{\partial r_r}{\partial t}\right|_{\text{evap}}}\right)$$
$$\times P(w,r_t,\theta_l,r_r,N_r)\,\mathrm{d}N_r\,\mathrm{d}r_r\,\mathrm{d}\theta_l\,\mathrm{d}r_t\,\mathrm{d}w. \tag{A25}$$

The PDF is transformed and a substitution is made that relates $S$ to $\chi$, as found in Larson and Griffin (2013). The equation becomes

$$\overline{w'\left.\frac{\partial r_r}{\partial t}\right|'_{\text{evap}}}$$
$$= \sum_{i=1}^{n}\xi_{(i)}\int\limits_{-\infty}^{\infty}\int\limits_{-\infty}^{\infty}\int\limits_{-\infty}^{\infty}\int\limits_{0}^{\infty}\int\limits_{0}^{\infty} (w-\overline{w})\left(C_{\text{evap}}\chi^{\alpha}\left(H(-\chi)\right)^{\alpha}r_r^{\beta}N_r^{\gamma} - \overline{\left.\frac{\partial r_r}{\partial t}\right|_{\text{evap}}}\right)$$
$$\times P_{(i)}(w,\chi,\eta,r_r,N_r)\,\mathrm{d}N_r\,\mathrm{d}r_r\,\mathrm{d}\eta\,\mathrm{d}\chi\,\mathrm{d}w. \tag{A26}$$

The coefficient $C_{\text{evap}}$ is given by

$$C_{\text{evap}} = 3\,c_{\text{evap}*}G\left(\overline{T_l},p\right)\left(\frac{4}{3}\pi\rho_l\right)^{\gamma}\left(\frac{1+\Lambda\left(\overline{T_l}\right)r_{sw}\left(\overline{T_l},p\right)}{r_{sw}\left(\overline{T_l},p\right)}\right)^{\alpha}, \tag{A27}$$

where $\rho_l$ is the density of liquid water and the function $G\left(\overline{T_l},p\right)$ is the coefficient in the drop radius growth equation (Rogers and Yau, 1989, Eq. 7.17). The constant $c_{\text{evap}*}$ is the ratio of raindrop mean geometric radius to raindrop mean volume radius, and is set by KK to a value of $0.86$. The variable $\eta$ is integrated out of the PDF, and the integral equation for the covariance of $w$ and evaporation rate is

$$\overline{w'\left.\frac{\partial r_r}{\partial t}\right|'_{\text{evap}}}$$
$$= C_{\text{evap}}\sum_{i=1}^{n}\xi_{(i)}$$
$$\times\left(f_{p(i)}\int\limits_{-\infty}^{\infty}\int\limits_{-\infty}^{\infty}\int\limits_{0}^{\infty}\int\limits_{0}^{\infty}(w-\overline{w})\left(\chi^{\alpha}\left(H(-\chi)\right)^{\alpha}r_r^{\beta}N_r^{\gamma} - \frac{1}{C_{\text{evap}}}\overline{\left.\frac{\partial r_r}{\partial t}\right|_{\text{evap}}}\right)\right.$$
$$\times P_{NNLL(i)}(w,\chi,r_r,N_r)\,\mathrm{d}N_r\,\mathrm{d}r_r\,\mathrm{d}\chi\,\mathrm{d}w$$
$$\left. -\left(1-f_{p(i)}\right)\left(\mu_{w(i)}-\overline{w}\right)\frac{1}{C_{\text{evap}}}\overline{\left.\frac{\partial r_r}{\partial t}\right|_{\text{evap}}}\right), \tag{A28}$$

where $P_{NNLL(i)}(w,\chi,r_r,N_r)$ is the $i$th component quadrivariate PDF involving two normal variates and two lognormal variates. The functional form of the PDF (for the $i$th PDF component) is given in the Supplement in Eq. (S1), and the integral is solved (for the $i$ PDF component) in Section S5 (Eq. (S9) through Eq. (S24)).

The integral equation for the covariance of $r_t$ and evaporation rate involves the PDF-variables $r_t$, $\theta_l$, $r_r$, and $N_r$. The equation is

$$\overline{r_t' \left.\frac{\partial r_r}{\partial t}\right|_{\text{evap}}}' = \int_{-\infty}^{\infty} \int_{-\infty}^{\infty} \int_{0}^{\infty} \int_{0}^{\infty} \left(r_t - \overline{r_t}\right) \left(\left.\frac{\partial r_r}{\partial t}\right|_{\text{evap}} - \overline{\left.\frac{\partial r_r}{\partial t}\right|_{\text{evap}}}\right)$$

$$\times P\left(r_t, \theta_l, r_r, N_r\right) \mathrm{d}N_r \,\mathrm{d}r_r \,\mathrm{d}\theta_l \,\mathrm{d}r_t. \tag{A29}$$

The PDF is transformed, and Eq. (A1) is used to substitute for $r_t$. The equation becomes

$$\overline{r_t' \left.\frac{\partial r_r}{\partial t}\right|_{\text{evap}}}' = \sum_{i=1}^{n} \xi_{(i)} \int_{-\infty}^{\infty} \int_{-\infty}^{\infty} \int_{0}^{\infty} \int_{0}^{\infty} \left(\mu_{r_t(i)} - \overline{r_t} + \frac{\left(\eta - \mu_{\eta(i)}\right) + \left(\chi - \mu_{\chi(i)}\right)}{2c_{r_t(i)}}\right)$$

$$\times \left(C_{\text{evap}} \chi^{\alpha} \left(H\left(-\chi\right)\right)^{\alpha} r_r^{\beta} N_r^{\gamma} - \overline{\left.\frac{\partial r_r}{\partial t}\right|_{\text{evap}}}\right)$$

$$\times P_{(i)}\left(\eta, \chi, r_r, N_r\right) \mathrm{d}N_r \,\mathrm{d}r_r \,\mathrm{d}\chi \,\mathrm{d}\eta. \tag{A30}$$

The covariance equation for $r_t$ and evaporation rate is split and simplified, resulting in

$$\overline{r_t' \left.\frac{\partial r_r}{\partial t}\right|_{\text{evap}}}'$$

$$= C_{\text{evap}} \sum_{i=1}^{n} \xi_{(i)} f_{p(i)}$$

$$\times \left(\frac{1}{2c_{r_t(i)}} \int_{-\infty}^{\infty} \int_{-\infty}^{\infty} \int_{0}^{\infty} \int_{0}^{\infty} \left(\eta - \mu_{\eta(i)}\right) \left(\chi^{\alpha} \left(H\left(-\chi\right)\right)^{\alpha} r_r^{\beta} N_r^{\gamma} - \frac{1}{C_{\text{evap}}} \overline{\left.\frac{\partial r_r}{\partial t}\right|_{\text{evap}}}\right)\right.$$

$$\times P_{NNLL(i)}\left(\eta, \chi, r_r, N_r\right) \mathrm{d}N_r \,\mathrm{d}r_r \,\mathrm{d}\chi \,\mathrm{d}\eta$$

$$+ \frac{1}{2c_{r_t(i)}} \int_{-\infty}^{0} \int_{0}^{\infty} \int_{0}^{\infty} \chi^{\alpha+1} r_r^{\beta} N_r^{\gamma} P_{NLL(i)}\left(\chi, r_r, N_r\right) \mathrm{d}N_r \,\mathrm{d}r_r \,\mathrm{d}\chi$$

$$+ \left.\left(\mu_{r_t(i)} - \overline{r_t} - \frac{\mu_{\chi(i)}}{2c_{r_t(i)}}\right) \int_{-\infty}^{0} \int_{0}^{\infty} \int_{0}^{\infty} \chi^{\alpha} r_r^{\beta} N_r^{\gamma} P_{NLL(i)}\left(\chi, r_r, N_r\right) \mathrm{d}N_r \,\mathrm{d}r_r \,\mathrm{d}\chi\right), \tag{A31}$$

where $P_{NLL(i)}\left(\chi, r_r, N_r\right)$ is the $i$th component trivariate PDF involving one normal variate and two lognormal variates. The functional form of the quadrivariate NNLL PDF (for the $i$th PDF component) is given in the Supplement in Eq. (S1), and the related integral is solved (for the $i$ PDF component) in Section S5 (Eq. (S9) through Eq. (S24)). The functional form of the trivariate NLL PDF (for the $i$th PDF component) is given in Eq. (S3), and the related integrals are solved (for the $i$th PDF component) by using the general form given in Section S7 (Eq. (S33) through Eq. (S40)).

The integral equation for the covariance of $\theta_l$ and evaporation rate involves the PDF-variables $r_t$, $\theta_l$, $r_r$, and $N_r$. The equation is

$$\overline{\theta_l' \left.\frac{\partial r_r}{\partial t}\right|_{\text{evap}}}' = \int_{-\infty}^{\infty} \int_{-\infty}^{\infty} \int_{0}^{\infty} \int_{0}^{\infty} \left(\theta_l - \overline{\theta_l}\right) \left(\left.\frac{\partial r_r}{\partial t}\right|_{\text{evap}} - \overline{\left.\frac{\partial r_r}{\partial t}\right|_{\text{evap}}}\right)$$

$$\times P\left(r_t, \theta_l, r_r, N_r\right) \mathrm{d}N_r \, \mathrm{d}r_r \, \mathrm{d}\theta_l \, \mathrm{d}r_t. \tag{A32}$$

The PDF is transformed, and Eq. (A2) is used to substitute for $\theta_l$. The equation becomes

$$\overline{\theta_l' \left.\frac{\partial r_r}{\partial t}\right|_{\text{evap}}}' = \sum_{i=1}^{n} \xi_{(i)} \int_{-\infty}^{\infty} \int_{-\infty}^{\infty} \int_{0}^{\infty} \int_{0}^{\infty} \left(\mu_{\theta_l(i)} - \overline{\theta_l} + \frac{\left(\eta - \mu_{\eta(i)}\right) - \left(\chi - \mu_{\chi(i)}\right)}{2c_{\theta_l(i)}}\right)$$

$$\times \left(C_{\text{evap}} \chi^{\alpha} \left(H\left(-\chi\right)\right)^{\alpha} r_r^{\beta} N_r^{\gamma} - \overline{\left.\frac{\partial r_r}{\partial t}\right|_{\text{evap}}}\right)$$

$$\times P_{(i)}\left(\eta, \chi, r_r, N_r\right) \mathrm{d}N_r \, \mathrm{d}r_r \, \mathrm{d}\chi \, \mathrm{d}\eta. \tag{A33}$$

The covariance equation for $\theta_l$ and evaporation rate is split and simplified, resulting in

$$\overline{\theta_l' \left.\frac{\partial r_r}{\partial t}\right|_{\text{evap}}}'$$

$$= C_{\text{evap}} \sum_{i=1}^{n} \xi_{(i)} f_{p(i)}$$

$$\times \left(\frac{1}{2c_{\theta_l(i)}} \int_{-\infty}^{\infty} \int_{-\infty}^{\infty} \int_{0}^{\infty} \int_{0}^{\infty} \left(\eta - \mu_{\eta(i)}\right) \left(\chi^{\alpha} \left(H\left(-\chi\right)\right)^{\alpha} r_r^{\beta} N_r^{\gamma} - \frac{1}{C_{\text{evap}}} \overline{\left.\frac{\partial r_r}{\partial t}\right|_{\text{evap}}}\right)\right.$$

$$\times P_{NNLL(i)}\left(\eta, \chi, r_r, N_r\right) \mathrm{d}N_r \, \mathrm{d}r_r \, \mathrm{d}\chi \, \mathrm{d}\eta$$

$$- \frac{1}{2c_{\theta_l(i)}} \int_{-\infty}^{0} \int_{0}^{\infty} \int_{0}^{\infty} \chi^{\alpha+1} r_r^{\beta} N_r^{\gamma} P_{NLL(i)}\left(\chi, r_r, N_r\right) \mathrm{d}N_r \, \mathrm{d}r_r \, \mathrm{d}\chi$$

$$+ \left.\left(\mu_{\theta_l(i)} - \overline{\theta_l} + \frac{\mu_{\chi(i)}}{2c_{\theta_l(i)}}\right) \int_{-\infty}^{0} \int_{0}^{\infty} \int_{0}^{\infty} \chi^{\alpha} r_r^{\beta} N_r^{\gamma} P_{NLL(i)}\left(\chi, r_r, N_r\right) \mathrm{d}N_r \, \mathrm{d}r_r \, \mathrm{d}\chi\right). \tag{A34}$$

The functional form of the quadrivariate NNLL PDF (for the $i$th PDF component) is given in the Supplement in Eq. (S1), and the related integral is solved (for the $i$ PDF component) in Section S5 (Eq. (S9) through Eq. (S24)). The functional form of

10    the trivariate NLL PDF (for the $i$th PDF component) is given in Eq. (S3), and the related integrals are solved (for the $i$th PDF component) by using the general form given in Section S7 (Eq. (S33) through Eq. (S40)).

**Code availability**

The CLUBB code is freely available for non-commercial use after registering for an account on the website http://clubb.larson-group.com. The specific version of CLUBB used in this paper is available in the SVN repository located at http://carson.math.uwm.edu/repos/clubb_repos/tags/MVCS.

5  *Acknowledgements.*  The authors thank three anonymous reviewers for the time they donated to reviewing the manuscript. The authors are grateful for financial support from the National Science Foundation under Grant No. AGS-0968640 and the Office of Science (BER), U. S. Department of Energy under Grant No. DE-SC0008323 (Scientific Discoveries through Advanced Computing, SciDAC). The large-eddy simulation presented here was performed on the Avi high-performance computer cluster at the University of Wisconsin – Milwaukee.

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

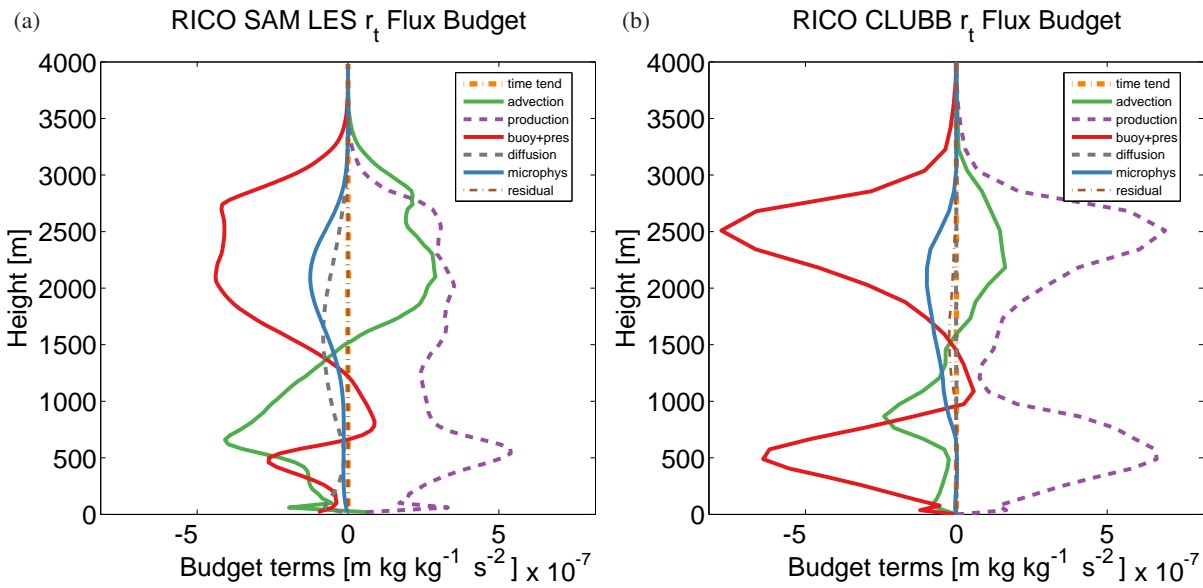

**Figure 1.** Profiles of $\overline{w'r_t'}$ budget terms for the RICO precipitating shallow cumulus case, time-averaged over the last half (36 hours) of the simulation (minutes 2160 through 4320), for (a) SAM LES and (b) CLUBB SCM. The profiles of overall time tendency are orange dashed-dotted lines, the advection terms are green solid lines, and the production terms are purple dashed lines. The sum of the buoyancy and pressure terms are the red solid lines. The diffusion terms are gray dashed lines, the microphysics (precipitation) terms are blue solid lines, and the residuals are brown dashed-dotted lines. SAM LES shows that the microphysics term is modest, but not negligible. The CLUBB microphysics term has the same sign and approximate magnitude as the SAM LES microphysics term.

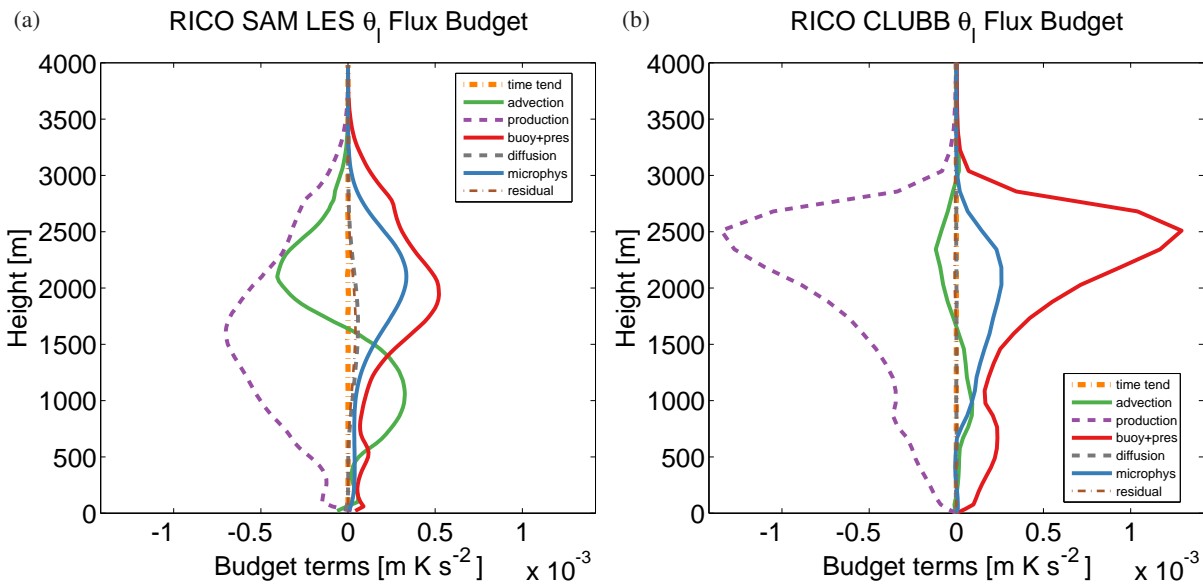

**Figure 2.** Profiles of $\overline{w'\theta_l'}$ budget terms for the RICO precipitating shallow cumulus case, time-averaged over the last half (36 hours) of the simulation (minutes 2160 through 4320), for (a) SAM LES and (b) CLUBB SCM. The profiles of overall time tendency are orange dashed-dotted lines, the advection terms are green solid lines, and the production terms are purple dashed lines. The sum of the buoyancy and pressure terms are the red solid lines. The diffusion terms are gray dashed lines, the microphysics (precipitation) terms are blue solid lines, and the residuals are brown dashed-dotted lines. SAM LES shows that the microphysics term is more significant for $\overline{w'\theta_l'}$ than it was for $\overline{w'r_t'}$. The CLUBB microphysics term has the same sign and approximate magnitude as the SAM LES microphysics term.

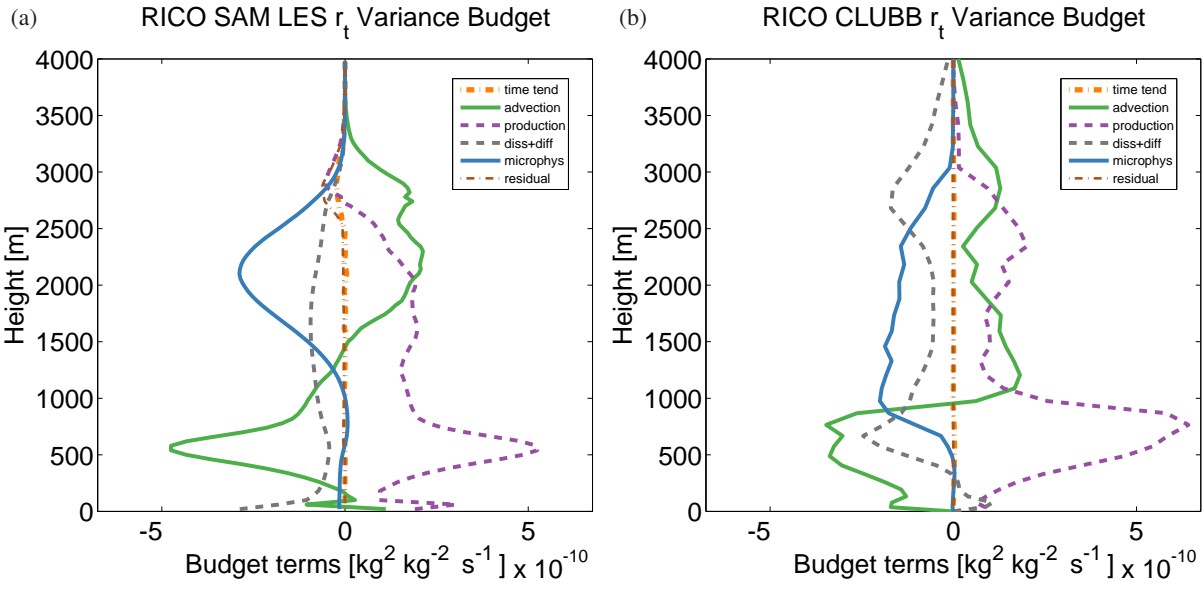

**Figure 3.** Profiles of $\overline{r_t'^2}$ budget terms for the RICO precipitating shallow cumulus case, time-averaged over the last half (36 hours) of the simulation (minutes 2160 through 4320), for (a) SAM LES and (b) CLUBB SCM. The profiles of overall time tendency are orange dashed-dotted lines, the advection terms are green solid lines, and the production terms are purple dashed lines. The sum of the dissipation and diffusion terms are gray dashed lines. The microphysics (precipitation) terms are blue solid lines, and the residuals are brown dashed-dotted lines. SAM LES shows that the microphysics term is significant. The CLUBB microphysics term is also significant, and has the same sign as the SAM LES microphysics term.

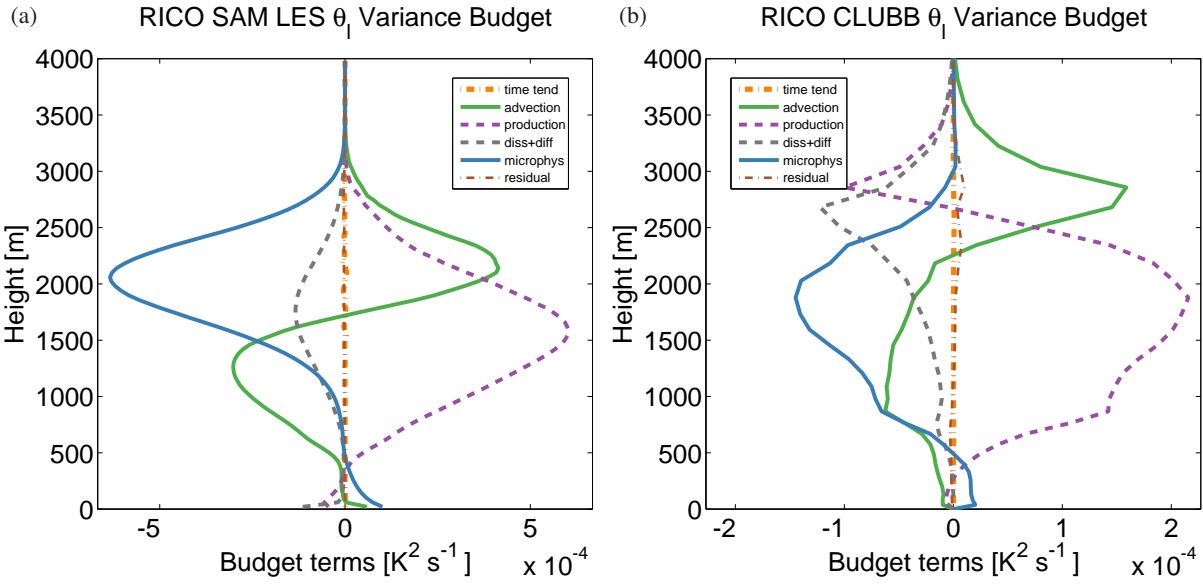

**Figure 4.** Profiles of $\overline{\theta_l'^2}$ budget terms for the RICO precipitating shallow cumulus case, time-averaged over the last half (36 hours) of the simulation (minutes 2160 through 4320), for (a) SAM LES and (b) CLUBB SCM. The profiles of overall time tendency are orange dashed-dotted lines, the advection terms are green solid lines, and the production terms are purple dashed lines. The sum of the dissipation and diffusion terms are gray dashed lines. The microphysics (precipitation) terms are blue solid lines, and the residuals are brown dashed-dotted lines. Note that the horizontal axes on the SAM LES and CLUBB panels are different. SAM LES shows that the microphysics term is a dominant sink term in the budget at cloudy levels, but then becomes a source of $\overline{\theta_l'^2}$ in the sub-cloud layer. The CLUBB microphysics term is also a dominant term at cloudy levels, balancing the production term, and also becomes a source of $\overline{\theta_l'^2}$ below cloud base.

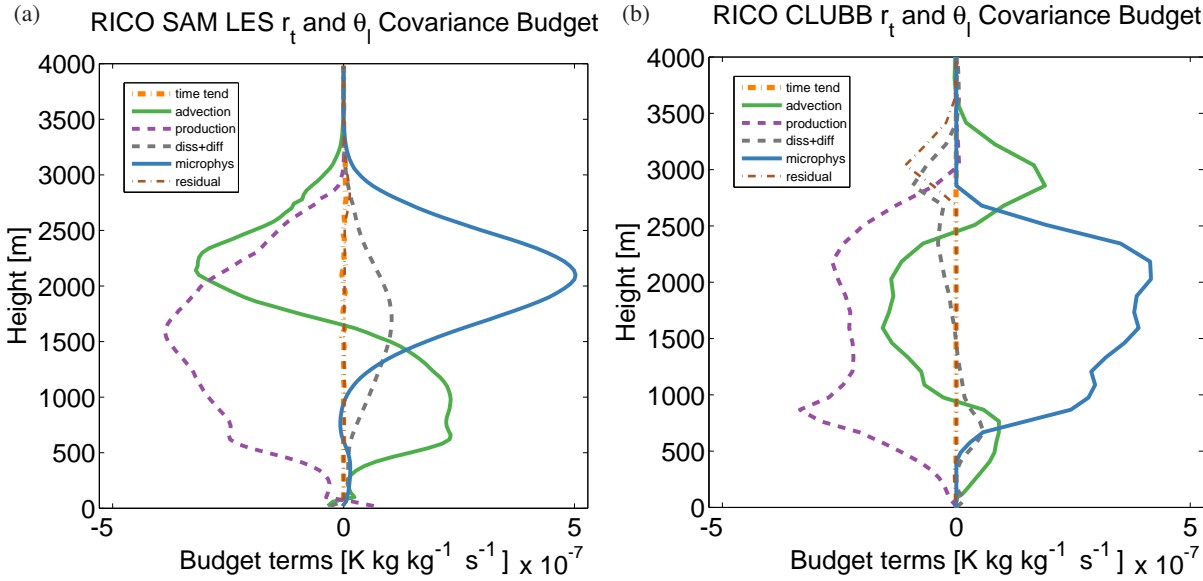

**Figure 5.** Profiles of $\overline{r'_t \theta'_l}$ budget terms for the RICO precipitating shallow cumulus case, time-averaged over the last half (36 hours) of the simulation (minutes 2160 through 4320), for (a) SAM LES and (b) CLUBB SCM. The profiles of overall time tendency are orange dashed-dotted lines, the advection terms are green solid lines, and the production terms are purple dashed lines. The sum of the dissipation and diffusion terms are gray dashed lines. The microphysics (precipitation) terms are blue solid lines, and the residuals are brown dashed-dotted lines. Again, SAM LES shows that the microphysics term is dominant. The CLUBB microphysics term is also dominant, and balances the production term in the budget.

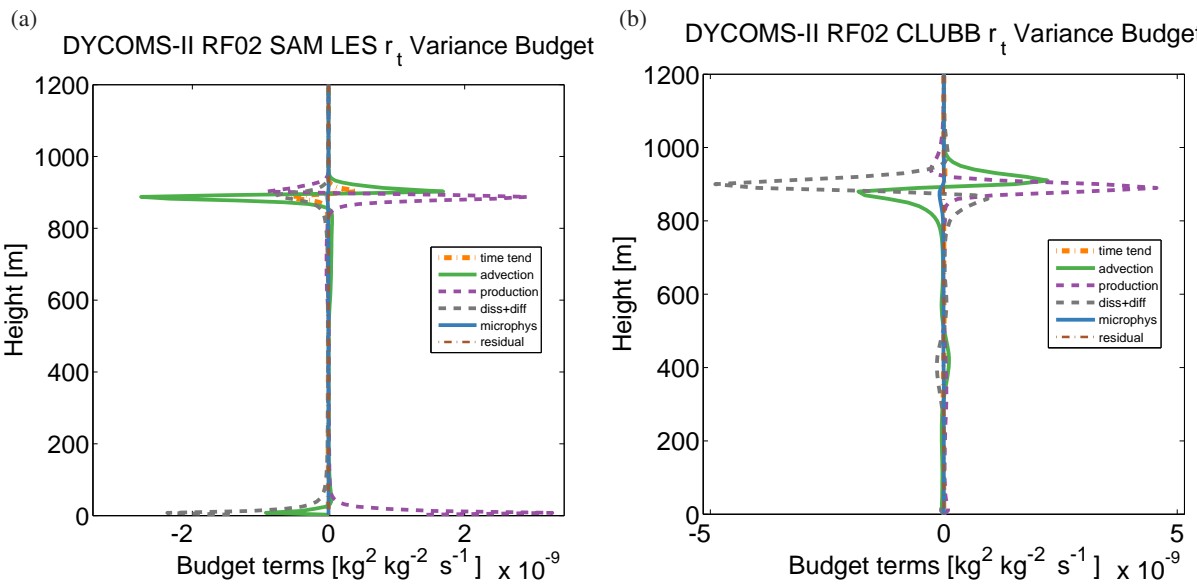

**Figure 6.** Profiles of $\overline{r_t'^2}$ budget terms for the DYCOMS-II RF02 drizzling stratocumulus cumulus case, time-averaged over the last hour (hour 6) of the simulation, for (a) SAM LES and (b) CLUBB SCM. The profiles of overall time tendency are orange dashed-dotted lines, the advection terms are green solid lines, and the production terms are purple dashed lines. The sum of the dissipation and diffusion terms are gray dashed lines. The microphysics (precipitation) terms are blue solid lines, and the residuals are brown dashed-dotted lines. Note that the the horizontal axis scale differs in the right and left panels. In this case, the microphysics term is negligible in both SAM LES and CLUBB. In this respect, the two models match, as desired.

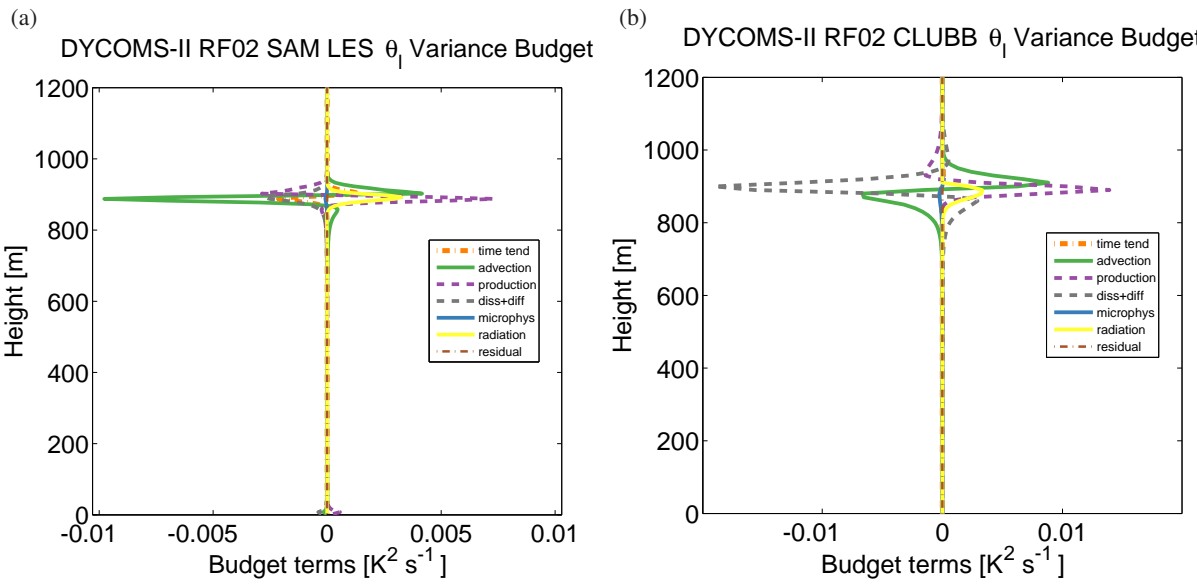

**Figure 7.** Profiles of $\overline{\theta_l'^2}$ budget terms for the DYCOMS-II RF02 drizzling stratocumulus case, time-averaged over the last hour (hour 6) of the simulation, for (a) SAM LES and (b) CLUBB SCM. The profiles of overall time tendency are orange dashed-dotted lines, the advection terms are green solid lines, and the production terms are purple dashed lines. The sum of the dissipation and diffusion terms are gray dashed lines. The microphysics (precipitation) terms are blue solid lines, the radiation terms are yellow solid lines, and the residuals are brown dashed-dotted lines. Note that the horizontal axes have different scales in the right and left panels. The microphysics term is negligible in both SAM LES and CLUBB. In this respect, both models match, as desired.

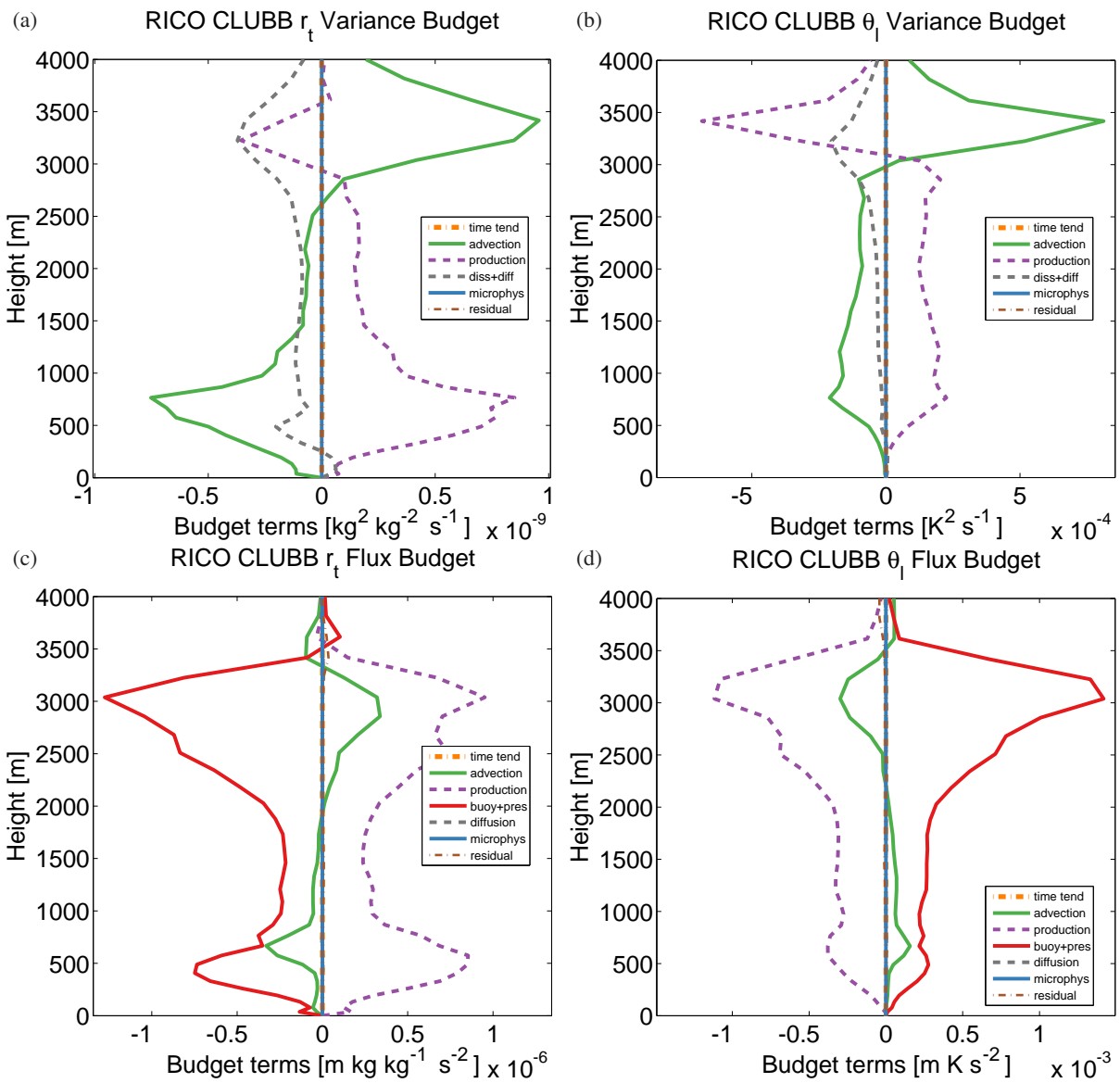

**Figure 8.** Profiles of budget terms for (a) $\overline{r_t'^2}$, (b) $\overline{\theta_l'^2}$, (c) $\overline{w'r_t'}$, and (d) $\overline{w'\theta_l'}$ for the RICO precipitating shallow cumulus case, time-averaged over the last half (36 hours) of the simulation (minutes 2160 through 4320), for CLUBB with the effects of microphysics on variances and covariances disabled. The profiles of overall time tendency are orange dashed-dotted lines, the advection terms are green solid lines, and the production terms are purple dashed lines. The sum of the buoyancy and pressure terms are the red solid lines. The diffusion (or the sum of diffusion and dissipation) terms are gray dashed lines, the microphysics (precipitation) terms are blue solid lines, and the residuals are brown dashed-dotted lines. Disabling the microphysical (co)variance terms greatly alters the budget balances for those fields. For the scalar variances (panels (a) and (b)), both dissipation and advection increase in magnitude.

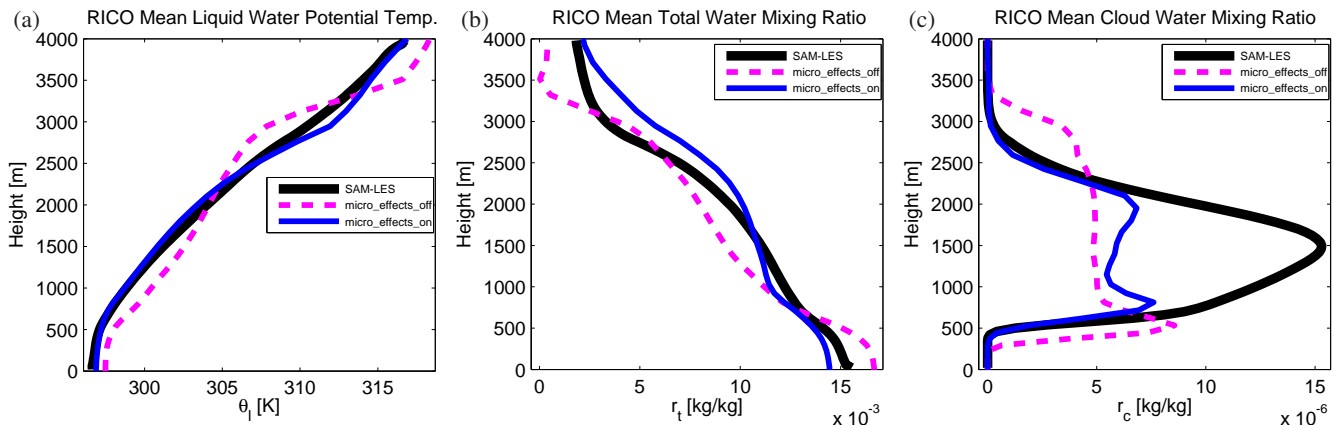

**Figure 9.** Profiles of (a) $\overline{\theta_l}$, (b) $\overline{r_t}$, and (c) $\overline{r_c}$ for the RICO precipitating shallow cumulus case, time-averaged over the last half (36 hours) of the simulation (minutes 2160 through 4320). The black solid lines are SAM LES results, the blue solid lines are CLUBB with the effects of microphysics on the variances and covariances ($\overline{w'r_t'}$, $\overline{w'\theta_l'}$, $\overline{r_t'^2}$, $\overline{\theta_l'^2}$, and $\overline{r_t'\theta_l'}$) enabled, and the magenta dashed lines are CLUBB with the effects of microphysics on the aforementioned variances and covariances turned off. Disabling the microphysical feedbacks into the (co)variances produces a $\overline{\theta_l}$ profile that is too warm at lower altitudes and too cool aloft when compared to SAM LES. This is because turning off the microphysical damping increases the vigor of the layer. As a result, cloud water is found at altitudes higher than it is found in SAM LES.