# Peer review of "Parameterizing microphysical effects on variances and covariances of moisture and heat content using a multivariate PDF: A study with CLUBB (tag MVCS)"

_Geoscientific Model Development, 2016_

## Short Comment (SC1) · 6 Jun 2016

Dear authors,

In my role as Executive editor of GMD, I would like to bring to your attention our Editorial version 1.1:

http://www.geosci-model-dev.net/8/3487/2015/gmd-8-3487-2015.html

This highlights some requirements of papers published in GMD, which is also available on the GMD website in the 'Manuscript Types' section:

http://www.geoscientific-model-development.net/submission/manuscript_types.html

In particular, please note that for your paper, the following requirements have not been met in the Discussions paper:

- "The main paper must give the model name and version number (or other unique identifier) in the title."

- "If the model development relates to a single model then the model name and the version number must be included in the title of the paper. If the main intention of an article is to make a general (i.e. model independent) statement about the usefulness of a new development, but the usefulness is shown with the help of one specific model, the model name and version number must be stated in the title. The title could have a form such as, "Title outlining amazing generic advance: a case study with Model XXX (version Y)"."

In the article the CLUBB model is named frequently. Therefore it should also be named in the title including a version number. Please correct this in your revised submission to GMD.

Yours,

Astrid Kerkweg

––––––––––––––––––––––––––––––

---

## Referee Comment (RC1) · Anonymous Referee #1 · 8 Jul 2016

The paper integrates various microphysical terms in higher-order moment equations analytically and compares the results with those from LES. It is overall a very boring paper. It is more like a technical report than a formal paper. If the readers do not need to dig into the model code, One does not think a reader can go through all the formulae.

The results are not new. Most of the them were documented in the literature. The differences between the analytical solution and LESs are kind of expected. More in-depth analysis may make the paper reach more readers and more interesting.

The method is not new, for example, Cheng and Xu (2009) published a pioneer work

in Journal of Atmospheric Science using the similar pdfs to integrate for various micro-physical terms. The analyses and experiments are more interesting than here. The author do not cite this work. This is not a good scholar.

Therefore, major revisions are needed.

––––––––––––––––––––––

---

## Referee Comment (RC2) · Anonymous Referee #2 · 13 Jul 2016

The authors compare an LES simulation to a single-column simulation with a model based on higher-order closure relations (CLUBB). Closure relations for the effects of microphysics on turbulent correlations are derived by integrating a simple microphysical model over assumed distribution functions for the fluctuating quantities.

The main use of this kind of comparison would be to improve the CLUBB model. Indeed, the figures show that there are large difference between the models at all heights. I was expecting the causes of these differences to be explored by sensitivity testing with CLUBB (or the LES) and concrete suggestions made for how the CLUBB could be improved to better represent the LES. There is a comment about this at the end section 4, with the suggestion that model improvements are 'out of scope', but without further exploration of the model differences the paper is somewhat dry and technical. I suggest that a major revision is necessary, to include an adequate exploration of the causes of the model differences.

I was expecting to see analytical results, somewhere in this paper, for the closed expressions for the microphysical covariances. The authors give expressions, e.g., Eq. (25), in terms of integrals, but do not actually evaluate the integrals in terms of the model prognostics. Given that it is the integrated expressions which are potentially useful to the reader, these should be given. This comment applies to the appendix as well.

The highly concise notation used in Section 2.2 for the PDFs is difficult to understand. Could it be replaced with one or two, well chosen, examples? If the fully-general expressions need to be recorded here, they could be moved to an appendix.

The figures comparing the LES and CLUBB often use different scales for the two models. This makes direct comparison difficult and should be corrected.

---

## Referee Comment (RC3) · Anonymous Referee #3 · 11 Aug 2016

This paper formulates relations for the effects of microphysical terms on turbulence variance and correlations using a multivariate PDF. Overall, I found the introductory, overview, and methodology sections to be quite well written while providing a clear motivation for the work. However, I felt the results section seemed incomplete and could be expanded upon a bit to make for a more convincing and satisfying read. Please see the below list on topics I would recommend be addressed before publication could be considered.

1) How do the mean state fields compare between LES and CLUBB with the new

parameterization? Can it adequately simulate the thermodynamic and cloud structure?

2) In relation to the above point, what is the effect of the new parameterization on the simulated cloud field? i.e. How does the new version of CLUBB compare to the control version of CLUBB? Providing this information will make the paper stronger by giving the reader a sense of the potential benefits of the new parameterization rather than just showing that the budget terms sort of match up.

3) Only one case is presented here. Oftentimes this is a warning sign that cherry-picking of the results or over-tuning of one case was the result. It would be nice if the authors somehow address this concern. Have the authors tested their parameterization on a stratocumulus over cumulus type of case?

4) Overall, the budgets from CLUBB reasonably match LES, although there are some instances where there are significant differences. I feel the authors brush this aside by saying improving these are "out of the scope" of the current work. I feel at minimum, a discussion should be included pertaining to potential reasons for these deficiencies and how improvements could be beneficial for an overall improved simulation of clouds (related to points 1 and 2).

---

## Author Comment (AC1) · 20 Aug 2016

**Response to Anonymous Referee #3**

In this response to the reviewer's comments, the reviewer's comments are italicized, and our responses are in roman font.

*This paper formulates relations for the effects of microphysical terms on turbulence variance and correlations using a multivariate PDF.*

Thank you for your review.

[Figure]
*Overall, I found the introductory, overview, and methodology sections to be quite well written while providing a clear motivation for the work. However, I felt the results section seemed incomplete and could be expanded upon a bit to make for a more convincing and satisfying read. Please see the below list on topics I would recommend be addressed before publication could be considered.*

More results have been added to the paper, as per the reviewer's suggestions below. The main result of the paper remains the formulas in the Appendix and Supplement that integrate over the PDF, and their implementation in code. These formulas are new, non-trivial to derive, and necessary for checking the convergence of numerical methods of obtaining these terms.

*1) How do the mean state fields compare between LES and CLUBB with the new parameterization? Can it adequately simulate the thermodynamic and cloud structure?*

Yes, CLUBB can adequately simulate the mean state fields. The revised manuscript now includes a new figure that shows mean state fields that are relevant to and influenced by the microphysical covariance terms.

*2) In relation to the above point, what is the effect of the new parameterization on the simulated cloud field? i.e. How does the new version of CLUBB compare to the control version of CLUBB? Providing this information will make the paper stronger by giving the reader a sense of the potential benefits of the new parameterization rather than just showing that the budget terms sort of match up.*

To assess the effects of the microphysical covariance terms, we have performed a sensitivity study in which those terms are shut off. A new section has been added to the manuscript in order to describe that sensitivity study. It turns out that, in the budgets, other terms compensate for the omission of the microphysical covariance terms, and the mean fields are significantly changed. In short, when the damping from the microphysical covariance terms is removed, the solution becomes overly vigorous.

*3) Only one case is presented here. Oftentimes this is a warning sign that cherry-picking of the results or over-tuning of one case was the result. It would be nice if the authors somehow address this concern. Have the authors tested their parameterization on a stratocumulus over cumulus type of case?*

Unfortunately, we do not have a precipitating stratocumulus-over-cumulus case. However, the revised manuscript now includes plots from the DYCOMSII-RF02 drizzling stratocumulus case. In that case, the microphysical covariance terms are small in both the LES and in CLUBB. It is an informative null case that indicates that CLUBB asymptotes to a reasonable solution in the stratocumulus limit.

*4) Overall, the budgets from CLUBB reasonably match LES, although there are some instances where there are significant differences. I feel the authors brush this aside by saying improving these are "out of the scope" of the current work. I feel at minimum, a discussion should be included pertaining to potential reasons for these deficiencies and how improvements could be beneficial for an overall improved simulation of clouds (related to points 1 and 2).*

As discussed in the response to reviewer 2, the main error in the microphysical terms is that they extend throughout the cloud layer, rather than being confined to the upper half of the cloud layer, as in LES. The revised version of the manuscript provides the following explanation:

"However, in CLUBB, the range of altitudes where the microphysics budget terms have significant values is shifted lower than in SAM LES. This occurs because $\overline{r_r}$ peaks at a lower altitude in CLUBB than in SAM LES. The lower-altitude peak in rain, in turn, occurs because there is too much evaporation near cloud top, as shown in Fig. 7a of Griffin and Larson (2016). As noted there, the excessive evaporation is caused by an excessively long-tailed marginal subgrid PDF of saturation deficit, which extends to unrealistically dry values. See Griffin and Larson (2016) for more details. The excessive evaporation near cloud top also causes a similar problem in the microphysical terms in

the other budgets presented below."

**References**

Griffin, B. M. and V. E. Larson, 2016: A new subgrid-scale representation of hydrometeor fields using a multivariate PDF. *Geosci. Model Dev.*, **9**, 2031–2053.

---

## Author Comment (AC2) · 20 Aug 2016

**Response to Anonymous Referee #2**

In this response to the reviewer's comments, the reviewer's comments are italicized, and our responses are in roman font.

*The authors compare an LES simulation to a single-column simulation with a model based on higher-order closure relations (CLUBB). Closure relations for the effects of microphysics on turbulent correlations are derived by integrating a simple microphysical model over assumed distribution functions for the fluctuating quantities.*

[Figure]

Thank you for your review.

*The main use of this kind of comparison would be to improve the CLUBB model.*

As stated in the original version of the manuscript, "A primary purpose of this paper is to perform those integrals analytically and to implement the resulting formulas in a particular PDF parameterization, the Cloud Layers Unified By Binormals (CLUBB) model." Performing the integrals involved considerable labor (see the Supplement), and, in our view, the resulting expressions and corresponding implementation in code are a substantial contribution in their own right. They are new, and they provide a benchmark solution for checking the convergence and accuracy of numerical methods for these terms.

The main use of comparing CLUBB results to LES is to give the reader an illustration of the size of the errors in the new microphysical covariance terms in a simple simulation, namely, the RICO shallow cumulus case.

*Indeed, the figures show that there are large difference between the models at all heights. I was expecting the causes of these differences to be explored by sensitivity testing with CLUBB (or the LES) and concrete suggestions made for how the CLUBB could be improved to better represent the LES. There is a comment about this at the end section 4, with the suggestion that model improvements are 'out of scope', but without further exploration of the model differences the paper is somewhat dry and technical. I suggest that a major revision is necessary, to include an adequate exploration of the causes of the model differences.*

The main error in the microphysical terms is that they extend throughout the cloud layer, rather than being confined to the upper half of the cloud layer, as in LES. The source of this error has been explored in a prior paper. The revised version of the manuscript provides the following explanation:

"However, in CLUBB, the range of altitudes where the microphysics budget terms have

significant values is shifted lower than in SAM LES. This occurs because $\overline{r_r}$ peaks at a lower altitude in CLUBB than in SAM LES. The lower-altitude peak in rain, in turn, occurs because there is too much evaporation near cloud top, as shown in Fig. 7a of Griffin and Larson (2016). As noted there, the excessive evaporation is caused by an excessively long-tailed marginal subgrid PDF of saturation deficit, which extends to unrealistically dry values. The excessive evaporation near cloud top also causes a similar problem in the microphysical terms in the other budgets presented below. See Griffin and Larson (2016) for more details."

*I was expecting to see analytical results, somewhere in this paper, for the closed expressions for the microphysical covariances. The authors give expressions, e.g., Eq. (25), in terms of integrals, but do not actually evaluate the integrals in terms of the model prognostics. Given that it is the integrated expressions which are potentially useful to the reader, these should be given. This comment applies to the appendix as well.*

The integrated expressions were and are contained in the Supplement, but that fact was not made explicit enough in the original text.

To clarify, the introduction now states "The needed integrals are set up in Appendix A and are solved by the expressions given in the Supplement."

In addition, the Appendix now notes that "This Appendix sets up the integrals that need to be solved in order to find the microphysical covariance terms listed in Section 2.1. The integrals set up here can be evaluated using the expressions given in the Supplement."

*The highly concise notation used in Section 2.2 for the PDFs is difficult to understand. Could it be replaced with one or two, well chosen, examples? If the fully-general expressions need to be recorded here, they could be moved to an appendix.*

The fully general expressions are necessary, but several illuminating examples are

listed in the Supplement. This is now pointed out just after the general form of the PDF is listed:

"Eq. 22 lists the general functional form for the subgrid PDF, but specific examples of marginals for a single mixture component are written out in the Supplement to this article. (A marginal PDF is the PDF that remains when one or more variates are integrated out.) These examples help provide intuition about the shape of the PDF. For instance, a univariate normal marginal of the PDF is written in Eq. (S7), and a univariate lognormal is written in Eq. (S8). A normal distribution is symmetric, extends from $(-\infty, +\infty)$, and has short tails. A lognormal distribution is useful for representing the distribution of a quantity such as rain mixing ratio. Such distributions are non-negative and often have a peak at low values and a long tail of larger values extending to the right. They are not well represented by normal distributions.

Also useful for gaining intuition are the bivariate marginals listed in Section S3 of the Supplement. A normal-normal bivariate form is listed in Eq. (S4), a lognormal-lognormal form is listed in Eq. (S6), and a hybrid normal-lognormal form is listed in Eq. (S5). Where a lognormal variate appears, the corresponding axis takes on only non-negative values and has a long tail. Which bivariate form is used depends on which functional forms are used to represent the variates of interest, e.g., rain mixing ratio (lognormal) or extended cloud water (normal)."

*The figures comparing the LES and CLUBB often use different scales for the two models. This makes direct comparison difficult and should be corrected.*

The scales for all but one of the RICO figures have been made the same. In order to make the individual budget terms more visible and distinguishable, the scales for the $\overline{\theta_l'^2}$ budget remain different, but that fact is now clearly noted in the appropriate figure caption:

"Note that the horizontal axes on the SAM LES and CLUBB panels are different."

**References**

Griffin, B. M. and V. E. Larson, 2016: A new subgrid-scale representation of hydrometeor fields using a multivariate PDF. *Geosci. Model Dev.*, **9**, 2031–2053.

---

## Author Comment (AC3) · 20 Aug 2016

**Response to Executive Editor**

*In the article the CLUBB model is named frequently. Therefore it should also be named in the title including a version number. Please correct this in your revised submission to GMD.*

The article is now entitled "Parameterizing microphysical effects on variances and co-variances of moisture and heat content using a multivariate PDF: A study with CLUBB (tag MVCS)".

---

## Author Comment (AC4) · 20 Aug 2016

**Response to Anonymous Referee #1**

In this response to the reviewer's comments, the reviewer's comments are italicized, and our responses are in roman font.

 *The paper integrates various microphysical terms in higher-order moment equations analytically and compares the results with those from LES.*

Thank you for your review.

*It is overall a very boring paper. It is more like a technical report than a formal paper. If the readers do not need to dig into the model code, One does not think a reader can go through all the formulae.*

Our paper is designed to satisfy the requirements of a "Model description paper" of GMD. Model description papers are a little different than typical scientific papers. The GMD webpage on manuscript types states: "Model description papers are comprehensive descriptions of numerical models which fall within the scope of GMD. The papers should be detailed, complete, rigorous . . . ideally, the description should be sufficiently detailed to in principle allow for the re-implementation of the model by others". The formulas listed in the paper must be included for completeness, rigor, and reproducibility.

One interesting aspect of our paper is the introduction of a technique to parameterize an effect of cold pools on convection, namely the increase in temperature variance below cloud. The revised manuscript emphasizes this important application of the method with sentences such as "The fact that CLUBB is able to parameterize this effect opens the door to future parameterization of the effects of cold pools on convection."

*The results are not new. Most of the them were documented in the literature.*

The paper's results are new. To our knowledge, no prior publication has parameterized microphysical effects on *variances and covariances.* Those terms are present in the governing equations, but all prior 1D parameterizations have omitted them. For instance, Cheng and Xu (2009) parameterized the effects of microphysics on grid means but not (co)variances. The microphysical effects on the grid means can shift the subgrid PDF to smaller or larger values, but they cannot directly change the *shape* of the PDF. In contrast, the covariance terms change the shape of the PDF. They are important because 1) they damp variability (i.e. narrow the PDF) via the effects of precipitation rather than turbulence; and 2) they generate variability (i.e. widen the PDF) below cloud via the effects of rain evaporation.

This is now clarified in the introduction.

*The differences between the analytical solution and LESs are kind of expected.*

The main error in the microphysical terms is that they extend throughout the cloud layer, rather than being confined to the upper half of the cloud layer, as in LES. The source of this error has been explored in a prior paper. The revised version of the manuscript provides the following explanation:

"However, in CLUBB, the range of altitudes where the microphysics budget terms have significant values is shifted lower than in SAM LES. This occurs because $\overline{r_r}$ peaks at a lower altitude in CLUBB than in SAM LES. The lower-altitude peak in rain, in turn, occurs because there is too much evaporation near cloud top, as shown in Fig. 7a of Griffin and Larson (2016). As noted there, the excessive evaporation is caused by an excessively long-tailed marginal subgrid PDF of saturation deficit, which extends to unrealistically dry values. The excessive evaporation near cloud top also causes a similar problem in the microphysical terms in the other budgets presented below. See Griffin and Larson (2016) for more details."

*More in-depth analysis may make the paper reach more readers and more interesting.*

To assess the effects of the microphysical covariance terms, we have performed a sensitivity study in which those terms are shut off. A new section has been added to the manuscript in order to describe that sensitivity study. It turns out that, in the budgets, other terms compensate for the omission of the microphysical covariance terms, and the mean fields are significantly changed. In short, when the damping from the microphysical covariance terms is removed, the solution becomes overly vigorous.

*The method is not new, for example, Cheng and Xu (2009) published a pioneer work in Journal of Atmospheric Science . . .*

Prior to Cheng and Xu (2009), several articles on the effects of microphysics on *grid means* were published, e.g., Zhang et al. (2002), Larson and Griffin (2006), and Morrison and Gettelman (2008).

*. . . using the similar pdfs to integrate for various micro- physical terms.*

The most important difference between the present paper and Zhang et al. (2002), Larson and Griffin (2006), Morrison and Gettelman (2008), and Cheng and Xu (2009) is that those papers omit the terms that are the central focus of the present paper: the effects of microphysics on *(co)variances*.

The effects on covariances are important terms in the covariance budgets, as demonstrated in the present paper's figures. But even when the effects on covariances are important, they have been neglected in prior papers. For instance, Eqn. (16) of Cheng and Xu (2009) prognoses the variance of rain water mixing ratio using an equation that includes turbulent advection, turbulent production, and turbulent dissipation, but not any effects of autoconversion, accretion, evaporation, or any other microphysical process.

The PDF shape used by Cheng and Xu (2009) for rain mixing ratio is a double Gaussian. The present paper uses a double lognormal. Using double lognormal has a key advantage for non-negative variables such as rain: a double lognormal never takes on negative values. For rain, this is important, because rain often has a small mean and a large right-hand tail at large values. Fitting the mean and variance of this shape using a double Gaussian will leave a significant fraction of the PDF to the left of zero, which is unphysical.

*The analyses and experiments are more interesting than here.*

The present paper includes complete budgets, whereas Cheng and Xu (2009) does not. Complete budgets are interesting because they allow the reader to compare the magnitude of the microphysical terms versus other terms, such as turbulence terms. For example, the budgets reveal the interesting fact that in our large-eddy simulation of a shallow cumulus case, microphysical damping of scalar variances is stronger than turbulent damping! This is important because the turbulent damping term is always included in scalar variance parameterizations, but microphysical damping has always

been neglected.

In addition, the revised manuscript includes the aforementioned sensitivity study in which the microphysical covariance terms are shut off.

*The author do not cite this work. This is not a good scholar.*

The original submission did not cite Cheng and Xu (2009), nor Zhang et al. (2002) and Larson and Griffin (2006), because those articles considered only with the effects of microphysics on grid means, not (co)variances.

The revised manuscript includes the following clarification and citations: "To clarify, we note that the microphysical terms we study here appear in the variance and covariance equations, not the grid mean equations. Microphysical effects on the grid means have been studied in several prior works (e.g., Zhang et al., 2002; Larson and Griffin, 2006; Morrison and Gettelman, 2008; Cheng and Xu, 2009; Larson and Griffin, 2013; Griffin and Larson, 2013; Boutle et al., 2014). The microphysical effects on the grid means can shift the subgrid Probability Density Function (PDF) to smaller or larger values, but, unlike the covariance terms, they cannot directly change the shape of the PDF. The microphysical covariance terms are important because 1) they damp variability (i.e. narrow the PDF) via the effects of precipitation rather than turbulence; and 2) they generate variability (i.e. widen the PDF) below cloud via the effects of rain evaporation."

*Therefore, major revisions are needed.*

The paper has been revised appropriately.

**References**

Boutle, I., S. Abel, P. Hill, and C. Morcrette, 2014: Spatial variability of liquid cloud and rain: Observations and microphysical effects. *Quart. J. Royal Met. Soc.*, **140**, 583–594.

Cheng, A. and K.-M. Xu, 2009: A pdf-based microphysics parameterization for simulation of drizzling boundary layer clouds. *J. Atmos. Sci.*, **66**, 2317–2334.

Griffin, B. M. and V. E. Larson, 2013: Analytic upscaling of local microphysics parameterizations, Part II: Simulations. *Quart. J. Royal Met. Soc.*, **139**, 58–69.

Griffin, B. M. and V. E. Larson, 2016: A new subgrid-scale representation of hydrometeor fields using a multivariate PDF. *Geosci. Model Dev.*, **9**, 2031–2053.

Larson, V. E. and B. M. Griffin, 2006: Coupling microphysics parameterizations to cloud parameterizations. Preprints*, 12th Conference on Cloud Physics*, Madison, WI, American Meteorological Society.

Larson, V. E. and B. M. Griffin, 2013: Analytic upscaling of local microphysics parameterizations, Part I: Derivation. *Quart. J. Royal Met. Soc.*, **139**, 46–57.

Morrison, H. and A. Gettelman, 2008: A new two-moment bulk stratiform cloud microphysics scheme in the Community Atmosphere Model, Version 3 (CAM3). Part I: Description and numerical tests. *J. Climate.*, **21**, 3642–3659.

Zhang, J., U. Lohmann, and B. Lin, 2002: A new statistically based autoconversion rate parameterization for use in large-scale models. *J. Geophys. Res.*, **107**, Article No. 4750. doi:10.1029/2001JD001484.